# Aerobic mechanochemical reversible-deactivation radical polymerization

Haoyang Feng[1], Zhe Chen[2], Lei Li[1], Xiaoyang Shao[1], Wenru Fan[1], Chen Wang[1], Lin Song [1], Krzysztof Matyjaszewski[3] ✉, Xiangcheng Pan [2] ✉ & Zhenhua Wang [1] ✉

Polymer materials suffer mechano-oxidative deterioration or degradation in the presence of molecular oxygen and mechanical forces. In contrast, aerobic biological activities combined with mechanical stimulus promote tissue regeneration and repair in various organs. A synthetic approach in which molecular oxygen and mechanical energy synergistically initiate polymerization will afford similar robustness in polymeric materials. Herein, aerobic mechanochemical reversible-deactivation radical polymerization was developed by the design of an organic mechano-labile initiator which converts oxygen into activators in response to ball milling, enabling the reaction to proceed in the air with low-energy input, operative simplicity, and the avoidance of potentially harmful organic solvents. In addition, this approach not only complements the existing methods to access well-defined polymers but also has been successfully employed for the controlled polymerization of (meth)acrylates, styrenic monomers and solid acrylamides as well as the synthesis of polymer/perovskite hybrids without solvent at room temperature which are inaccessible by other means.

Aerobic physical activities relying on molecular oxygen and mechanical exercises could promote tissue regeneration and repair in various organs primarily mediated by stem cells and progenitor cells in skeletal muscle, nervous system, and vascular system[1–3]. However, synthetic systems are constantly challenged by oxygen and external mechanical forces, giving rise to the mechano-oxidative deterioration or degradation of polymer materials by rupturing covalent bonds in the backbones of polymers[4–6]. Although various (macro)molecular engineering approaches have replicated parts of the biological aerobic activities, including oxygen tolerance[7–11] or mechanical adaptivity[12–17], none of them can synergistically replicate the combinative attributes of molecular oxygen and mechanical forces via a productive pathway.

Reversible-deactivation radical polymerization (RDRP) mediated by the chemical equilibrium between active and dormant species has enabled excellent control over the macromolecular chain structure[18–20]. Recent advances[21–33] in mechanochemical radical polymerization have further extended the possibility of RDRP to the mechano-responsive systems, including heterogenous curing gels[31,32], self-growing polymers[34,35], and self-strengthened materials[36,37]. Mechanically controlled radical polymerization relying on piezo-electricity, contact electrification or sonolysis require metal-based mechanotransducers or high-energy force to activate polymerization via mechano-electro-chemical transformation, constraining their applications, particularly in biomedicine and electronics. Furthermore, as polymerization could be terminated by molecular oxygen, most of these elegant designs were required to be conducted under anaerobic conditions. Recently, enormous effort has been devoted to the removal of dissolved oxygen prior to polymerization, including approaches employing enzymes[8,9,38], microbial metabolisms[7,10,11], reducing agents[39–43], or photocatalysts[44–46]. The broad success of

[1]Frontiers Science Center for Flexible Electronics (FSCFE) & Institute of Flexible Electronics (IFE), Northwestern Polytechnical University, Xi'an 710072, China. [2]State Key Laboratory of Molecular Engineering of Polymers, Department of Macromolecular Science, Fudan University, Shanghai 200438, China. [3]Department of Chemistry, Carnegie Mellon University, 4400 Fifth Avenue, Pittsburgh, PA 15213, USA. ✉e-mail: matyjaszewski@cmu.edu; panxc@fudan.edu.cn; iamzhwang@nwpu.edu.cn

## Aerobic Mechanochemical RDRP based on organic mechano-labile complexes

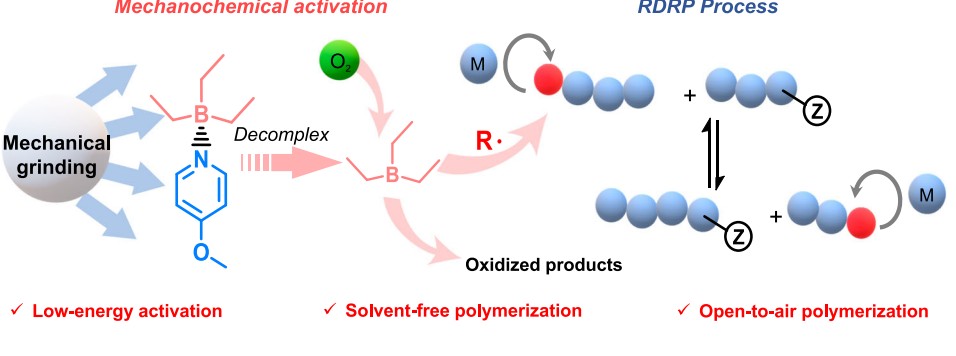

**Fig. 1 | Illustrative design of aerobic mechano-RDRP.** Aerobic mechano-RDRP process dismantles into mechanochemical activation and RDRP process two parts. Organic mechano-labile complexes (molecules labeled red and blue) were activated by mechanical grinding (white ball) and then decomplex. Reactive radicals (red R) are generated from the oxidative process of complexes in active state (molecules labeled red). Reactive radicals initiate the polymerization of vinyl monomers (blue ball and red ball) and mechanical grinding assists solvent-free RDRP process for efficient chain transfer (blue ball chain with Z group).

oxygen-tolerant RDRP hinges on the susceptibility of the chemical conversion of molecular oxygen with high levels of efficiency and selectivity[47]. A much more viable and ambitious solution to this grand challenge would be the development of a metal-free, oxygen-tolerant, and low-energy-input mechanochemical system for RDRP.

During aerobic excises, muscle glycogen particles are broken down under external force, freeing glucose molecules that can be further oxidized through aerobic processes to produce the adenosine triphosphate (ATP) molecules required for biological activities[48–50]. Inspired by the unique profile of the aerobic process, we hypothesized that the regeneration of activators from molecular oxygen could be achieved through a mechanistically distinct approach using mechanical energy. Herein, we designed an organic mechano-labile initiator that could be activated by ball milling to release reactive species required for the oxidative process to produce initiating radicals for the polymerization of vinyl monomers to generate well-defined polymers (Fig. 1). This method of polymerizing vinyl monomers using ball milling not only features open-to-air reaction, low-energy input, operative simplicity, and the avoidance of potentially harmful organic solvents but also provides us with the unique opportunity to utilize molecular oxygen and applied stress for the controlled polymerization of solid monomers and the bulk synthesis of polymer/perovskite hybrids which are inaccessible by other means.

## Results

### Aerobic mechano-RAFT driven by ball-milling

The generation of activating adenosine triphosphate (ATP) throughout aerobic exercise relies on the coupling of mechanochemical activation of glycogen and oxidation of glucose[51]. Inspired by this, we designed a triethylborane/pyridine (Et$_3$B-Py) complex as the latent initiator consisting of triethylborane and an electron-donating pyridine for the mechanochemical conversion of oxygen into activators to enable RDRP. This design arose from previous findings that triethylborane could react with molecular oxygen in air to form initiating radicals for polymerization[39,52,53]. The reactivity of triethylborane was expected to be blocked by the judicious selection of ligand, then restored in response to ball milling (Fig. 2a). To verify the feasibility of this concept, we initially investigated the aerobic mechanochemical reversible addition-fragmentation chain transfer (mechano-RAFT) polymerization as the model method for the polymerization of n-butyl acrylate (nBA) as monomer, 2-butylsulfanyl-thiocarbonylsulfanyl-propionic acid (BTPA) as the chain transfer agent (CTA), and triethylborane/pyridine complex (Et$_3$B-PyOMe) as

the mechano-labile molecules (Fig. 2a). The pristine transparent liquids switched to transparent and sticky liquids after 2 h ball milling with 30 Hz oscillation frequency (Fig. 2b, up), and the $^1$H NMR spectroscopy revealed 80% conversion for the polymerization (Fig. 2c and Supplementary Fig. 1). The reactions without ball milling, air or initiator showed little conversion (<5%, Supplementary Fig. 2). To evaluate the influence of heat generated during ball milling, the temperature inside the milling jar throughout the polymerization was measured by thermography, increasing from 20.9 °C in the beginning to 27.9 °C after reaction (Fig. 2b, down). A control experiment at 30 °C was conducted, no signal of polymer could be observed based on the $^1$H NMR spectroscopy and GPC trace for the reaction (Fig. 2c and Supplementary Figs. 3 and 4). These results indicated that the polymerization was attributed to mechanochemical activation.

To further confirm that the polymerization is associated with mechanical force, reactions under various milling frequencies were conducted. Linear evolution of $M_n$ with conversion was observed for all the reactions. The length of the polymer chain steadily grew at 30 Hz, exhibiting the highest polymerization rate (Fig. 2d, e, and Supplementary Figs. 5 and 6), indicating the efficient and constant conversion of oxygen into initiating radicals under ball milling. With a lower frequency (10 Hz), an obvious induction period was observed, as displayed in Supplementary Figs. 7 and 8. Đ ($M_w/M_n$) values slightly decreased throughout all the polymerizations, as expected from a controlled process, with the final Đ being as low as 1.10 (Supplementary Figs. 9 and 10). Other potential factors, such as the dosage of the initiator (Supplementary Table 1), solvents for liquid-assisted grinding (LAG) (Supplementary Table 2), the structure of CTAs (Supplementary Table 3), milling balls (Supplementary Table 4), temperature (Supplementary Table 5), and air volume (Supplementary Table 6) were screened out to optimize reaction conditions.

To explore whether this low-energy-input approach could be used to synthesize polymers with a high degree of polymerization (DP) compared to the previous mechanochemical approach. We first explored the effect of target DP ($DP_T = 100/200/400$). All the reactions achieved high monomer conversion (>70%, Supplementary Fig. 11), giving well-defined polymers with excellent match between experimental molecular weight and theoretical value as well as low dispersity (Fig. 2f). Furthermore, we tried to synthesize a polymer with $DP_T = 1000$ and obtained a PnBA with an actual DP = 950 and Đ = 1.29 through the optimized method (Supplementary Fig. 12, details in "Methods" section). The slightly high dispersity in this case is

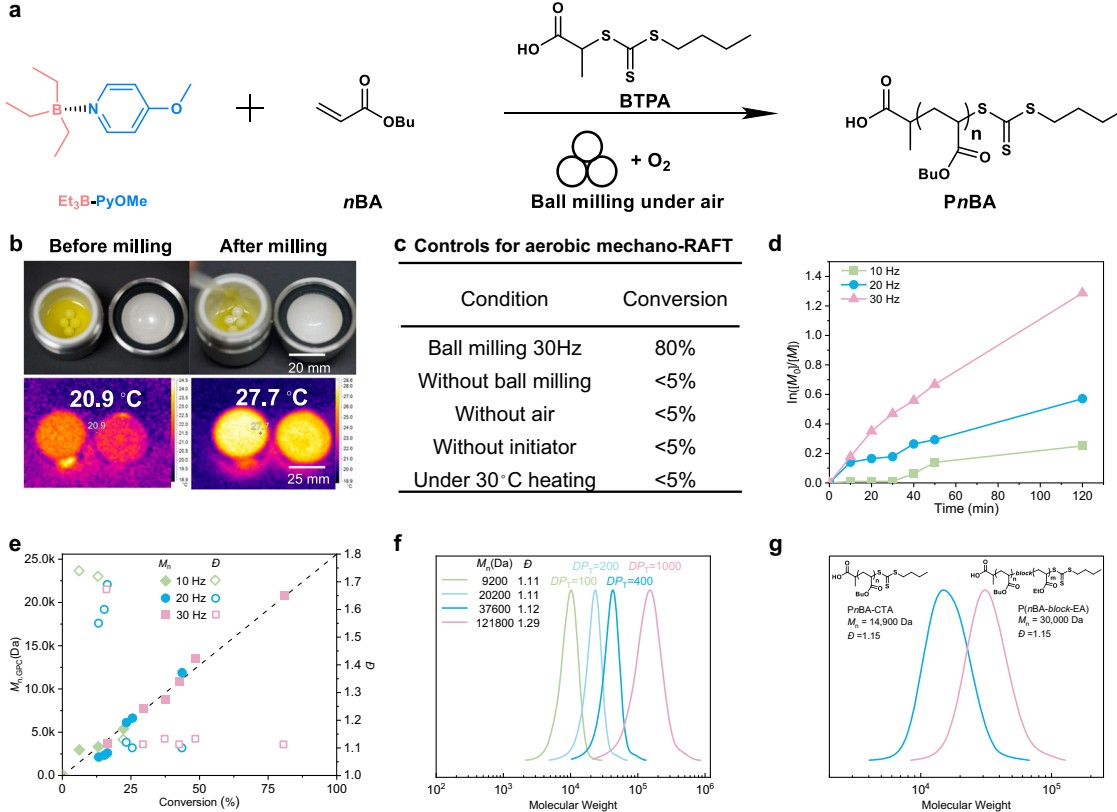

**Fig. 2 | Aerobic mechano-RAFT driven by ball-milling. a** Schematic polymerization of *n*BA. **b** Physical pictures and infrared thermal images of the ball milling jar before and after the reaction. **c** Controls for aerobic mechano-RAFT reaction conditions: [*n*BA]: [BTPA]: [Et₃B-PyOMe] = 200:1:5, 100 μL (0.03 mL/g) DMF as LAG, 35 mL zirconium oxide milling jar with four 8 mm diameter zirconium oxide balls, Reaction condition: 30 Hz-2 h, and conversion was determined by ¹H NMR spectroscopy. **d** Kinetic plot evolution of mechano-RAFT under various ball milling frequencies. **e** The evolution of molecular weight and dispersity for the polymerization versus conversion. **f** GPC traces of aerobic mechano-RAFT with different $DP_T$. **g** GPC traces of P*n*BA and chain-extended polymer.

potentially attributed to the presence of some oxygen, which may terminated some polymer chains, or a relatively high concentration of radicals[47].

Another feature of this aerobic mechano-RAFT is the high retention of chain end for the synthesized polymer. Matrix-assisted laser desorption/ionization-time of flight (MALDI-TOF) mass spectra of P*n*BA showed that the strongest peak revealed a molecular weight of 8054 Da (DP = 61, Supplementary Fig. 13), which was close to the corresponding GPC trace ($M_n$ = 7500 Da, Supplementary Fig. 14). Another distribution at 8069 Da can be attributed to the removal of a methyl group by laser irradiation[33], and the main peak with intervals of 128.03 Da corresponded to the molar mass of the *n*BA unit (Supplementary Fig. 15). Chain extension was further conducted to examine the chain-end fidelity of the synthesized polymer by this approach. *n*BA was polymerized under identical conditions to give the first block P*n*BA ($M_n$ = 14,900 Da and Đ = 1.15). Ethyl acrylate was used as the second monomer for the synthesis of the block copolymer (Supplementary Fig. 16). After chain extension, a clear shift to high-molecular-weight region ($M_n$ = 30,100 Da) and a narrow molecular weight distribution (Đ = 1.15) were observed in GPC traces (Fig. 2g), suggesting the high chain-end fidelity of P*n*BA synthesized via the aerobic mechano-RAFT polymerization.

## Mechanistic study of mechanochemical activation

Organic compounds with labile bonds have been demonstrated to be directly activated by external force, including ball milling[54]. To elucidate the mechanochemical activation of Et₃B-PyOMe, molecular electrostatic potential (MESP) was simulated to monitor the structural evolution of Et₃B-PyOMe before and after dissociation. As shown in Fig. 3a, the triethylborane fragment in complex manifested negative electrostatic potentials (blue color), and the pyridine fragment exhibited positive electrostatic potentials (red color). After decomplexation, triethylborane was transformed to positive electrostatic potentials (white color) due to the electron-deficient character of the boron atom, and the nitrogen atom in the pyridine ligand performed obvious negative electrostatic potentials (blue color). Fourier transform infrared (FTIR) spectroscopy was further employed to analyze the chemical environment of the pyridine ligand. Along with ball milling, the breathing (at around 1595 cm⁻¹) and stretching peak (at around 760 cm⁻¹) of the pyridine ring with Et₃B-PyOMe was significantly enhanced (Fig. 3b and Supplementary Fig. 17)[55], revealing an obvious overlap with that of free PyOMe. Taking Et₃B as the control, there was no signal in the band around 760 cm⁻¹ before and after oxidation. These results indicated that the decomplexation of Et₃B-PyOMe occurred during ball milling[56]. In addition, ¹¹B NMR spectroscopy was utilized to study the key boron intermediates (Fig. 3c). Initially, triethylborane (1 M in THF) was easily oxidized in air for 1 day, giving rise to the oxidized products including Et₂BOEt and (EtO)₂BEt. In the case of Et₃B-PyOMe, the ¹¹B NMR spectroscopy remained unchanged with a clear chemical shift (η) at 0.39 ppm after exposing to air for 1 week, indicating the stability of the complex under ambient conditions. We then subjected Et₃B-PyOMe to ball milling (20 Hz) under air atmosphere for 60 min. The spectra showed that most Et₃B-PyOMe (-97.2%) was transformed into oxidized products, including (EtO)₂BEt (-78.5%) and (EtO)₃B (-18.7%). To confirm the radical mechanism for the mechanochemical process, 2,2,6,6-tetramethylpiperidinyl-1-oxide (TEMPO) and 5,5-dimethyl-1-pyrroline n-oxide (DMPO) were respectively used to capture the generated ethyl

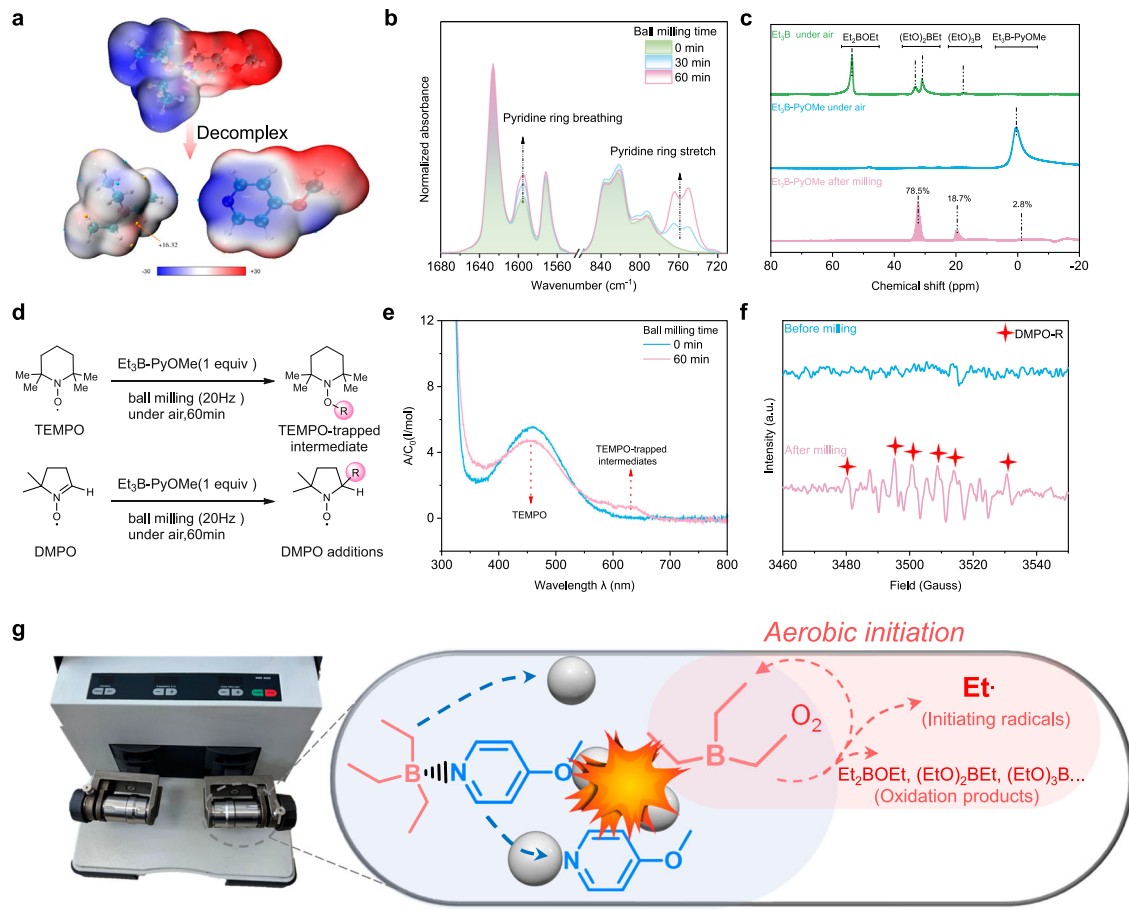

**Fig. 3 | Mechanistic analysis for mechanochemical initiation. a** MESP analysis of Et₃B-PyOMe before and after milling. **b** Infrared spectra of Et₃B-PyOMe under ball milling reaction (20 Hz). **c** ¹¹B NMR spectra (in CDCl₃) of triethylborane in air, Et₃B-PyOMe in air and after 20 Hz ball milling. **d** Radical capture route in ball milling. **e** Monitoring of TEMPO-trapped intermediates by UV−vis spectra. **f** EPR spectra of DMPO before and after ball milling. **g** Hypothetical process for the radicals produced by mechanical force and oxygen.

radicals (Fig. 3d). In the ultraviolet−visible light (UV−vis) spectra, an absorbance band centered at 469.5 nm for the pristine TEMPO was observed and decreased as ball milling (20 Hz) proceeded. A new absorbance band centered at 630 nm appeared as the TEMPO-trapped intermediates (Fig. 3e)[24,57]. Furthermore, the EPR spectra revealed that carbon-centered radicals were generated via the oxidative reaction of triethylborane and rapidly captured by DMPO to give stable adducts ($A_N$ = 15.10 G, and $A_{Hβ}$ = 21.50 G, Fig. 3f)[58,59]. Given by this, the mechanism of aerobic mechanochemical activation was outlined in Fig. 3g, comprising the mechanochemical release of triethylborane and oxidative reaction with oxygen to generate initiating radicals.

**Monomer scope for aerobic mechano-RAFT**

The monomer scope of this approach was extended to a variety of vinyl monomers, including solid monomers, as depicted in Table 1. The polymerization of (methy)acrylates, including *n*-butyl acrylate (*n*BA), ethyl acrylate (EA), tert-butyl acrylate (*t*BA), methyl acrylate (MA) and methyl methacrylate (MMA) was initially attempted under procedure 1 (at Table 1, Fig. 4). Well-defined polymers were achieved for all the acrylates, but the polymerization of MMA revealed a wider molecular weight distribution (*Đ* = 1.50) due to the slow initiating rate of BTPA for methacrylates (Supplementary Figs. 18−20). Then, 4-(benzenecarbonothioylsulfanyl)-4-cyanopentanoic acid (BTCPA) was selected as the CTA for the polymerization of MMA with 10 mm balls and an extended reaction time (5 h), afford a higher conversion

(~52%, Supplementary Fig. 21) and well-defined PMMA ($M_n$ = 8,300 Da, *Đ* = 1.19, Supplementary Fig. 22). In addition, the polymerization of styrene (St) and vinyl naphthalene (VN)) and vinyl carbazole (VC) was also performed (at Table 1, procedure 2 of Fig. 4). All the reactions produced well-defined polymers with predetermined molecular weight and narrow molecular weight

**Table 1 | Monomer scope for aerobic mechano-RAFT**

| Entry | Monomer | $DP_T$ | *Conv.* (%)[a] | $M_{n,th}$ (Da)[b] | $M_{n,GPC}$(Da)[c] | *Đ*[c] | *I*\* (%)[d] |
|---|---|---|---|---|---|---|---|
| 1 | *n*BA | 200 | 81 | 21,000 | 20,800 | 1.11 | >99 |
| 2 | EA | 200 | 75 | 15,200 | 13,400 | 1.14 | >99 |
| 3 | *t*BA | 200 | 65 | 16,900 | 12,600 | 1.16 | >99 |
| 4 | MA | 200 | 65 | 11,400 | 15,100 | 1.09 | 76 |
| 5 | MMA | 200 | 32 | 6600 | 8900 | 1.50 | 74 |
| 6 | St | 200 | 32 | 7100 | 6100 | 1.22 | >99 |
| 7 | VN | 100 | 30 | 4800 | 4400 | 1.21 | >99 |
| 8 | VC | 100 | 25 | 5050 | 5400 | 1.45 | >94 |
| 9 | NIPAM | 100 | >95 | 11,600 | 25,800 | 1.36 | 45 |
| 10 | NPA | 100 | >95 | 15,000 | 18,900 | 1.19 | 79 |

[a]Conversion was determined by ¹H NMR spectroscopy.
[b]$M_{n,theo}$ = $M_{end\ group}$ + [M]₀/[CTA]₀ × conversion × $M_{monomer}$.
[c]$M_n$ and $M_w/M_n$ were determined by GPC.
[d]Initiator efficiency (*I*\*) = $M_{n,theo}/M_{n,exp}$ × 100.

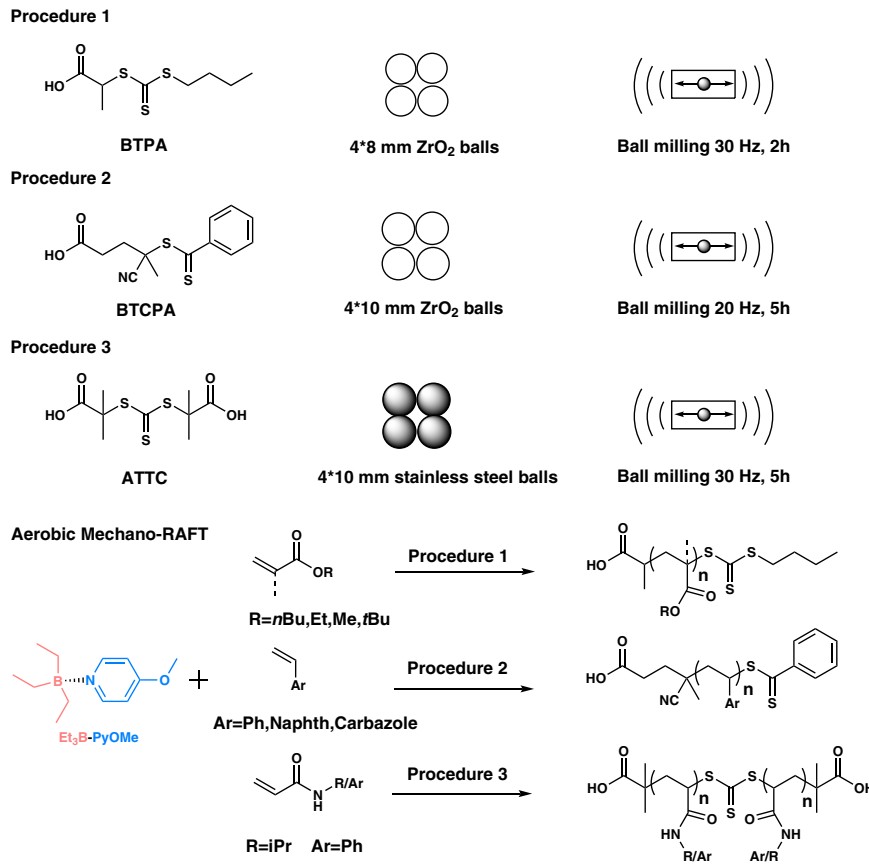

**Fig. 4 | Procedures for aerobic mechano-RAFT polymerization regarding various monomers.** Procedure conditions mainly included RAFT agents, grinding balls, oscillating frequency and time. These procedures are artificially programmed to adapt to the physical state and polymerization activity of various monomers ((meth)acrylates, styrenic monomers and acrylamides).

distribution (Supplementary Figs. 23–26), indicating that the aerobic mechano-RAFT could be employed for the polymerization of low-activity and solid monomers at room temperature. We further tried to polymerize solid acrylamide monomers using this mechanochemical approach to circumvent the phase transition of polyacrylamide during thermal polymerization[60,61]. Procedure 3, S,S'-bis(R,R'-dimethyl-R''-acetic acid)-trithiocarbonate (ATTC) as the CTA coupled with stainless steel jar and balls, was performed for the polymerization of solid acrylamide monomers. The polymerization for N-isopropyl acrylamide (NIPAM) and N-phenylacrylamide (NPA) monomers achieved nearly complete conversion (no monomer signal, Supplementary Figs. 27 and 28) and narrow molecular weight distribution (at Table 1, Supplementary Fig. 29) after 5 h (30 Hz with 4*10 mm stainless steel balls, procedure 3 of Fig. 4). It is essential to explore the upper molecular-weight limit of this approach as ball milling with stainless steel jar will generate strong mechanical force, giving rise to mechanochemical degradation of polymers. The GPC traces revealed a single peak with dispersities in the range of 1.14–1.52 with increasing the molar ratio of monomer to CTA from 50 to 200 (Supplementary Fig. 30). While in the case of $DP_T = 500$, a bimodal peak was observed in the GPC trace, indicating that the polymerization was out of control due to the high-energy mechanical input and highly viscous reaction.

## Hybrid synthesis based on aerobic mechano-RAFT

Unlike previous piezoelectrical[27,31–33] and triboelectrical[62] systems depending on a high loading of inorganic powders, this aerobic mechano-RAFT adopted a tiny amount of organic mechano-labile initiators. Thus, we envisioned this approach could be used for the bottom-up synthesis of polymer/inorganic hybrids without solvent and degassing. The synthesis of polymer/perovskites hybrids was attempted because polymer/perovskites hybrids have received intensive interest on account of their role in light-emitting diodes[63], photovoltaics[64], sensors[65–67], and thin-film transistors[68]. With this in mind, the synthesis of polymer/perovskites hybrids was attempted. nBA was used as the first monomer due to the excellent control of polymerization. NIPAM was selected as the functional monomer to coordinate and passivate perovskite nanocrystals (PNCs)[69]. As illustrated in Fig. 5a, PNCs@P(nBA-co-NIPAM) were synthesized by one-pot ball milling (20 Hz with four 10 mm zirconia balls) without additional solvent and further purification. Well-defined polymers were achieved with high conversion (>80%, Supplementary Fig. 31) and narrow molecular weight distribution ($Đ = 1.25$, Supplementary Fig. 32). The signal intensity analysis of $^{13}C$ NMR spectroscopy (DEPT135, $CH_3$/CH positive and $CH_2$ negative) shows that the copolymerization ratio of nBA and NIPAM segment is 16.9, which closed with the feeding ratio (Fig. 5b and Supplementary Fig. 33). The photoluminescence (PL) spectrum of this hybrids was displayed in Fig. 5c, with a maximum emissive wavelength around 529 nm and a full width at half-maximum (FWHM) of ~25.2 nm. In addition, the XRD pattern of the hybrid matched the standard card of $MAPbBr_3$ nanocrystals (Fig. 5d)[70]. "IFE" characters on the glass were made by the injection of PNCs @ P(nBA-co-NIPAM), affording strong green fluorescence emission under 365 nm ultraviolet irradiation (Fig. 5e). These results demonstrated the exciting potential of this aerobic mechano-RAFT as an operationally simple, mild and solvent-free route to synthesize polymer/inorganic hybrids from commercial monomers and precursors.

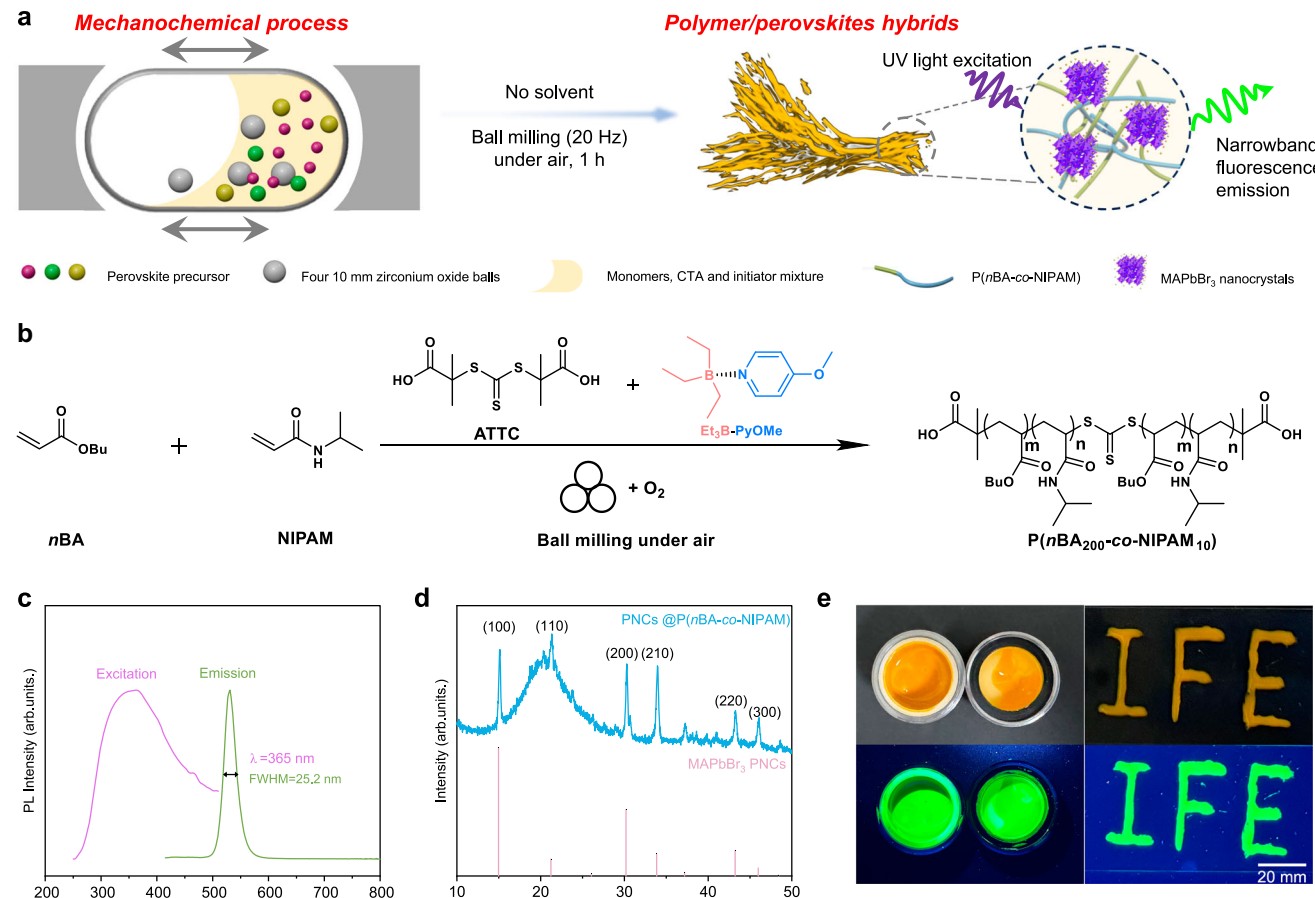

**Fig. 5 | Polymer/perovskites hybrids by aerobic mechano-RAFT. a** A brief schematic illustration of the synthesis of polymer/perovskites hybrids. **b** Copolymerization of *n*BA and NIPAM involved in the synthesis of hybrids. **c** PL absorbance spectrum with excitation wavelengths and PL emission spectrum under excitation at 365 nm. **d** XRD pattern of uniform luminescent hybrids (black line) and the vertical red sticks represent the peak positions and their respective relative intensities of MAPbBr$_3$ PNCs. **e** Physical pictures of the ball milling jar and IFE characters under natural light and under 365 nm ultraviolet lamp.

## Discussion

In summary, molecular oxygen and mechanical forces have been synergistically utilized to conduct aerobic mechanochemical reversible-deactivation radical polymerization by the deliberate design of an Et$_3$B/pyridine complex as the mechano-labile initiator that could release free Et$_3$B upon ball milling. The released Et$_3$B could further react with molecular oxygen to generate active radicals to induce solventless RAFT polymerization of a wide array of monomers, including (meth)acrylates, styrene monomers, and solid acrylamides at room temperature, with excellent control over chain length, dispersity, and high chain-end fidelity. This method enables the reaction to proceed in air with low-energy input, operative simplicity, and the avoidance of potentially harmful organic solvents. In addition, this approach not only complements the existing mechanochemical approaches for the control of macromolecular structure but also accesses well-defined polymer/perovskite hybrids without solvent, which are inaccessible by other mechanochemical means.

## Methods

### General aerobic mechano-RAFT procedure

Prior to polymerization, the ambient temperature was kept at $20 \pm 2\,°C$ to reduce the effect of temperature on polymerization. Afterwards, a general polymerization reaction mixture was prepared as follows. In a 35 mL zirconium oxide jar equipped with four 8 mm zirconium oxide balls, 3.6 mL (25.2 mM, 200 equiv.) butyl acrylate monomer, 0.030 g (0.126 mM, 1 equiv.) BTPA, 0.130 g (0.63 mM, 5 equiv.) Et$_3$B-PyOMe and 100 µL DMF (0.03 mL/g) were mixed. The final reaction mixtures were ball-milled at 30 Hz for 2 h.

### Optimized aerobic mechano-RAFT procedure for polymerization with $DP_T = 1000$

In 35 mL zirconium oxide jar equipped with four 8 mm zirconium oxide balls, 7.2 mL (50.4 mM, 1000 equiv.) butyl acrylate monomer, 0.012 g (0.050 mM, 1 equiv.) BTPA, 0.016 g (0.076 mM, 3 equiv.) Et$_3$B-PyOMe and 200 µL DMF (0.03 mL/g) were mixed. The final reaction mixtures were ball-milled at 30 Hz for 2 h.

### Chain extension

The macromolecular chain transfer agents (Macro-CTAs) were obtained from the general aerobic mechano-RAFT procedure (30 Hz, 1 h) of butyl acrylate monomer. In 35 mL zirconium oxide jar equipped with four 10 mm zirconium oxide balls, 2.56 g (20.0 mM, 200 equiv.) ethyl acrylate monomer, 1.49 g (0.10 mM, 1 equiv.) Macro-CTAs, 0.041 g (0.20 mM, 2 equiv.) Et$_3$B-PyOMe and 120 µL DMF (0.03 mL/g) were mixed. The final reaction mixtures were ball-milled at 30 Hz for 3 h.

### Optimized aerobic mechano-RAFT procedure 1

In 25 mL zirconium oxide jar equipped with four 8 mm zirconium oxide balls, 3.6 mL (25.2 mM, 200 equiv.) butyl acrylate monomer, 0.030 g (0.126 mM, 1 equiv.) BTPA, 0.130 g (0.63 mM, 5 equiv.) Et$_3$B-PyOMe and 100 µL (0.03 mL/g) DMF were mixed. The final reaction mixtures were ball-milled at 30 Hz for 2 h.

## Optimized aerobic mechano-RAFT procedure 2

In 25 mL zirconium oxide jar equipped with four 10 mm zirconium oxide balls, 2.62 g (25.2 mM, 200 equiv.) Styrene, 0.028 g (0.126 mM, 1 equiv.) CTBPA, 0.130 g (0.63 mM, 5 equiv.) $Et_3B$-PyOMe and 100 μL (0.03 mL/g) DMF were mixed. The final reaction mixtures were ball-milled at 20 Hz for 5 h.

## Optimized aerobic mechano-RAFT procedure 3

In a 25 mL stainless stell jar equipped with four 10 mm stainless stell balls, 2.85 g (25.2 mM, 100 equiv.) NIPAM, 0.071 g (0.252 mM, 1 equiv.) ATTC, and 0.157 g (0.756 mM, 3 equiv.) $Et_3B$-PyOMe were mixed. The final reaction mixtures were ball-milled at 30 Hz for 5 h.

## Synthesis of polymer/perovskites hybrids

In a 25 mL zirconium oxide jar equipped with four 10 mm zirconium oxide balls, 3.60 mL (25.2 mM, 200 equiv.) butyl acrylate monomer, 0.143 g (1.26 mM, 10 equiv.) NIPAM monomer, 0.034 g (0.126 mM, 1 equiv.) ATTC, 0.130 g (0.63 mM, 5 equiv.) $Et_3B$-PyOMe and 0.096 g (3 wt.%) perovskite precursor (MABr and $PbBr_2$ with molar ratio = 1:1) were mixed. The final reaction mixtures were ball-milled under air at 20 Hz for 1 h to obtain the luminescent composite with PNCs.

## NMR spectra of polymers

All the monomer conversions were measured by $^1H$ NMR in $CDCl_3$ of DMSO-$d_6$ using Bruker Avance Neo 500 MHz spectrometer at 25 °C. $^{13}C$ NMR spectra (DEPT 135, $CH_3$/CH positive, and $CH_2$ negative) were used to confirm the actual ratio of different monomer fragments in copolymerization.

## GPC traces of polymers

The molecular weights ($M_n$) and Đ ($M_w/M_n$) were determined by Gel Permeation Chromatography (GPC). The GPC was performed in tetrahydrofuran solution at 35 °C with an elution rate of 1.0 mL/min on an Agilent 1260 HPLC system equipped with a G7110B pump and a G7162A refractive index detector. while the entry of NIPAM and NPA monomers were used with DMF as the eluent at a flow rate of 1 mL/min at 35 °C. The apparent molecular weights were determined on a single PL gel MIXED-C columns using linear poly (methyl methacrylate) standards.

## MALDI-TOF mass spectroscopy of P*n*BA

PnBA for MALDI-TOF mass spectroscopy was obtained from a general aerobic mechano-RAFT procedure (30 Hz, 1 h). MALDI-TOF mass spectrometer was from Bruke, Germany. The MALDI instrument was equipped with a 337 nm pulsed nitrogen laser (laser intensity of 50 Hz). The number of laser irradiations was 100 for all mass spectra (delay time of 190 ns), with a 20 kV acceleration voltage. MALDI experiment was carried out using 2,5-DHB as the matrix. The matrix solution was prepared by dissolving 40 mg of 2,5-DHB in 1 mL of THF.

## UV−vis spectra of radical scavenger experiments with TEMPO

UV-vis spectra were measured using a U-3900H spectrophotometer (Hitachi, Japan). The concentration of TEMPO in a reaction mixture can be determined using the Beer−Lambert law ($A = \varepsilon bc$) by monitoring the characteristic UV−vis signal of TEMPO at 469.5 nm. In this equation, $A$ is the absorbance at 469.5 nm in the UV−vis spectrum, $\varepsilon$ (L/(mol cm)) is the molar absorptivity of TEMPO in DCM, $c$ (mol/L) is the concentration of TEMPO, and $b$ (cm) is the path length of the sample holder. The $\varepsilon b$ was considered as a constant in the same experiment. Therefore, a variation of the Beer−Lambert law ($A/c_0 = \varepsilon bc/c_0$) was used to measure the concentration change of TEMPO. A 0.20 mmol of DMPO dissolved in 10 mL DCM and 5 mL was taken out of it as a control. The remaining 5 mL was loaded into a 35 mL zirconium oxide milling jar with four 8 mm diameter zirconium oxide balls. Then, 0.10 mmol of $Et_3B$-PyOMe was added into the jar and milled for 30 or 60 min at 20 Hz.

## Experimental procedure for DMPO trapping and EPR measurements

1.00 mmol of DMPO dissolved in 5 mL DMSO and loaded into a 35 mL zirconium oxide milling jar with four 8-mm diameter zirconium oxide balls. Then 1.00 mmol of $Et_3B$-PyOMe was added into the jar and milled for 60 min at 20 Hz. The reaction mixture was collected directly with 0.5 mm capillary and analyzed using EPR. Control experiment: The $Et_3B$-PyOMe was not added, and the other operations were the same as those described above. EPR spectra were recorded at room temperature on the EPR Bruker EMXplus spectrometer operated at 9.855 GHz. Typical spectrometer parameters were shown as follows, sweep width: 100.00 G; center field set: 3505.00 G; conversion time: 2 ms; sweep time: 60 s; modulation amplitude: 1.0 G; modulation frequency: 100 kHz; PowerAtten:25.0 dB; microwave power: 0.6325 mW.

## Crystal structure and optical properties of polymer/perovskites hybrids

The crystal structure of polymer/perovskite hybrids was characterized using an X-ray diffractometer (Bruker AXS D8) equipped with a Cu-Kα radiation source. Steady-state and time-resolved PL spectra were measured on FLS1000 from Edinburgh Instruments with excitation wavelengths at 365 nm.

## Data availability

The synthesis methods, optimization studies, NMR spectra, GPC traces, and mass spectrometry data generated in this study are provided in the Supplementary Information/Source Data file. Other data used in this study are available in the figshare database under accession code https://doi.org/10.6084/m9.figshare.26131858. Data can also be obtained from the corresponding author upon request. Source data are provided in this paper.

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

## Acknowledgements

We thank the support from the National Natural Science Foundation of China (22322103 and 22271057 to X.P.), Natural Science Foundation of Shanghai (22ZR1406000 to X.P.)), and Shaanxi Province Natural Science Basic Research Program (2024JC-YBMS-294 to Z.W.). K.M. acknowledges support by NSF (DMR 2202747 to K.M.). We would like to acknowledge Dourong Wang, Guichen Liu, and Chengyu Fu for their fruitful discussions.

## Author contributions

Z.W. and X.P. conceived the idea and designed the experiments. H.F. conducted all polymerization as well as the vast majority of the characterization, including NMR, GPC, UV–vis, EPR, FTIR, FLS, and XRD in collaboration with X.S. and C.W. H.F. and X.S. conducted the synthesis of organic molecules in collaboration with Z.C. L.L. conducted MESP. H.F. conducted MALDI-ToF-MS analysis and interpreted the data in collaboration with W.F. H.F. conducted the hybrid synthesis in collaboration with L.S. H.F., X.S., and Z.W. analyzed all data with input from H.F. and X.S. H.F. and Z.W. co-wrote the paper. All authors commented on the paper. The overall supervision of the project was conducted by Z.W. and K.M.

## Competing interests

The authors declare no competing interests.
