## [Peer Review File · Nature Communications]

Aerobic Mechanochemical Reversible-Deactivation Radical PolymerizationReviewers' Comments:

Reviewer #1:

Remarks to the Author:

In this work, the authors present a latent triethylborane-pyridine complex as a mechanically activated source of radicals. Based on prior work from Pan (co author), triethylborane reduces molecular oxygen to a "radical species" that is capable of initiating polymerization processes.

This is a significant advancement in the field as it allows for the production of polyacrylates (and related materials) under milling conditions. Milling is "green" but also, as the authors demonstrate, provides a means by which to access materials that can't be made using conventional thermal/photochemical approaches (more on that later). I have a few major questions/concerns that need to be addressed before I can recommend for publications:

1. The authors constantly allude to "radical" species but don't actually define what is doing the initiation. More importantly, from a mechanistic standpoint, how does this species compare to what the active initiating species (singular or plural) is/are in solution (from their ACIE 2018 work)? Does solution-state vs. milling change the nature of these species? I think this type of mechanistic question is important for understanding how this process works and the ability to probe (mass spec, for example) differentiates this work from work that I might find in an applied polymer journal.
2. In Fig 2d, the authors claim they TEMPO trap a "radical" species. The authors use a solid arrow but don't provide a yield nor do they provide any experimental evidence of what is formed (aside from UV vis data). You can't claim to have made a new chemical species (solid arrow) without more rigorous characterization. This point closely relates to my concern above (#1) in that in this paper the authors don't provide any evidence of how chain growth kicks off.
3. I appreciate the polymer/perovskite application but the explanation about what the expected results are is insufficient/confusing. I assume these perovskites emit at 529 nm when well dispersed? But if they cluster they don't emit (photo-RDRP)? And if they sort of disperse but don't crystallize (thermal-RDRP) they also don't emit? This section seems to be not as clear as the rest of the manuscript. I suggest clarifying how all of the controls were designed, what expected results are, and rationale for each piece of data. The cartoons in Fig 4 are even more confusing. Where does the crosslinker come from? The authors haven't made crosslinked materials at all in the paper and then all of a sudden throw it into this experiment. The cartoons make it seem like perovskite and nanocrystal (I am confused about how these are different materials) aren't added into the vessels either. The SI provides a brief prep that suggests that nanocrystals ARE added to the milling jars but unfortunately there is no prep for the control experiments so I can not comment how these were performed. There needs to be some clarity here to help the reader follow what is going on. This section really detracts from an otherwise very nice manuscript.

Minor Points:

1. The IR spectral shifts described in the main text p4 should be qualified with specific structural features.
2. Do the authors know the conversion to three significant figures as suggested in Fig 3 (and in line text)?
3. Mn values in the main text should have associated units

Reviewer #2:

Remarks to the Author:

The team described ball-milling RAFT with aerobic initiation. It is clearly stated that ball-milling truly modulated RAFT, but the lack of details blurred the nice achievement. Even it presented new hybrid material fabrication, it is still not enough to convince this development truly expands a realm of synthetic polymer chemistry. Sorry, I can not recommend it for Nature Communications.

1. The identity of the initiation process remains unclear. Reference 24 detailed the chemical mechanism of Et₃B and oxygen at room temperature. The authors introduced pyridine to inhibit Et₃B, which is believed to be reactivated during ball-milling. Several spectroscopic pieces of evidence support Et₃B liberation and oxygen activation, but the structure of oxidized Et₃B-Py remains elusive. At the very least, endgroup analysis is necessary to comprehend the initiation process.
2. The use of DFT may not be appropriate. The authors applied directional force to break the coordination between Et₃B and pyridine, which I disagree with. These directional forces might be applicable to chemical bonds embedded in polymer chains. Many studies by Moore, Otsuka, Peterson, Yoon, and others suggest that sufficient chain length is necessary to stretch the chain and break the labile chemical bond in the middle of the chain. When short chains are used, no directional chemical deformation is observed with ball-milling. Therefore, DFT calculations of the small complex, Et₃B:Py, are not properly designed.
3. Polymerization without oxygen should be conducted to confirm oxygen's involvement. Additionally, control of chain length by TTC loading is necessary.
4. The scope of the monomers used is narrow. Only liquid monomers were used to achieve homogeneous mixing of monomer, initiator, and CTA, and all the examples are easily accessible through simple liquid polymerization. As mechanochemistry gains popularity, it is essential to explore the types of new polymers that can be synthesized through ball-milling. Previous examples by Bielawski (ATRP), Golder (RDRP), Borchardt (Polyphenylene), and Kim (ROMP) beautifully demonstrated the uniqueness of this approach compared to solution-based methods. While the formation of hybrid materials is commendable, it would be suitable for Nature Communications if you could demonstrate a broader monomer scope.
5. It is a well-known fact that ball-milling can lead to the degradation of high molecular-weight polymers. All the examples in Table 1 showed products with low molecular weights, which may be stable under ball-milling conditions. It would be beneficial to pursue the synthesis of high molecular-weight products, preferably exceeding 50 kDa.

Reviewer #3:

Remarks to the Author:

Feng et al. present in this manuscript a novel approach for conducting reversible addition-fragmentation chain transfer (RAFT) polymerization. Rather than utilizing a traditional thermal radical initiator, the authors utilized a triethylborane/pyridine (Et₃B-Py) complex and molecular oxygen as a radical source. Specifically, Et₃B released from the Et₃B-Py complex reacts with oxygen to generate radicals.

This manuscript brings to my mind a paper by Fedorov and colleagues from over a decade ago, in which a similar alkylborane complex was used to initiate free radical polymerization (*Macromolecules* 2007, 40, 10, 3554-3560). Additionally, the Pan group has also leveraged the reaction between triethylborane and oxygen to facilitate similar RAFT polymerization (*Angew. Chem.* 2018, 130, 9574-9577). Overall, the use of alkylboranes in conventional and controlled radical polymerization is well-documented in the literature (see this review: *J. Polym. Sci.* 2020, 58, 14-19). Therefore, the novelty and significance of this work in comparison to existing literature are limited.

I acknowledge that in this study, ball milling was employed to release Et₃B from the complex, whereas Fedorov and colleagues used light. Furthermore, the authors also applied this new approach to synthesize a polymer/perovskite hybrid material with luminescent properties. However, I would argue that such hybrid materials can be readily obtained through similar RAFT polymerization methods (*ACS Energy Lett.* 2022, 7, 2, 610-616) or numerous other techniques (refer to the excellent review: *Adv. Mater.* 2021, 33, 2005888). This work presents an alternative approach rather than the sole method to

synthesize the hybrid materials.

Altogether, while this work introduces a new approach to RAFT polymerization, its novelty and significance are relatively low. Therefore, I recommend publishing this work in a specialized polymer journal. The following recommendations aim to rectify flaws in data interpretation and enhance the quality of the manuscript.

1. This approach is based on RAFT polymerization and therefore, the absence of any mention of RAFT polymerization in the entire manuscript is a significant oversight. I strongly recommend that the authors add a paragraph about RAFT polymerization in the introduction and discuss this polymerization technique throughout the paper. T
2. The term Reversible-Deactivation Radical Polymerization (RDRP) includes RAFT, ATRP, NMP, etc., so it is overly general to use in the title, abstract, and main text. Does this approach also apply to ATRP and NMP?
3. On page 4, the authors conclude that "These results suggested that ball milling could change the binding state of pyridine ligands in the Et3B-Py complex and induce the oxidation of Et3B." This statement appears to be incorrect. Does the ball milling or molecular oxygen induce the oxidation of Et3B?
4. It is widely recognized that ball milling can disrupt bulk complex structures, increase surface area, and enhance reactivity. However, there is no scientific evidence to support the claim that ball milling can induce stretching and bond dissociation as depicted in Figure 2e. How can ball milling simulate external force at two specific locations as shown in Figure 2e? This DFT simulation is not convincing evidence to explain the bond dissociation and release of Et3B.
5. In Figure 2b, I recommend adding a control ¹¹B NMR spectroscopy of oxidized product without using ball milling (Et3B reacts with oxygen).
6. The comparison of aerobic mechano-RDRP in this work and skeletal muscle in aerobic exercise in Figure 1 and the introduction is not scientifically relevant. Aerobic physical activities can positively impact tissue regeneration through the release of growth factors, improvement of blood flow, and overall enhancement of tissue microenvironments. During aerobic exercise, the body supplies energy to skeletal muscles through various pathways, including those that do not always require oxygen such as Glycolysis. Additionally, the force applied during aerobic exercise does not aid in energy production by consuming oxygen, as in this chemistry approach, where the force from ball milling helps release Et3B and produce radicals. Such a comparison makes the work sound like a nature-mimicking system, but it is not accurate. I suggest removing this comparison.
7. The chain extension data in Figure S24 shows very little shift (from 26000 to 30000), which is not convincing evidence for high end group fidelity. The authors should aim to achieve a larger increase in molecular weight.
8. What is the kinetics of Et3B release/free from the complex? How does this release kinetics relate to the kinetics of radical formation and polymerization?
9. The comparison of photo- and thermo-RAFT (Figure 4) without using the ball milling is also not convincing. The uniformity and luminescent properties may be a result of the well-known mixing effect of the ball milling. If this is the case, why not just use traditional RAFT polymerization with a good mixing approach such as ball milling, high-shear flow or ultrasonification? I recommend the authors perform control experiments of photo- and thermo-RAFT (Figure 4) using the ball milling.

We warmly welcome the reviewers' comments and deeply appreciate the fact they have taken the time to examine the manuscript in great detail and provided many pertinent comments. The manuscript has been greatly improved following this round of review; we believe we have addressed the comments and concerns of the reviewers appropriately.

REVIEWER COMMENTS

Reviewer #1 (Remarks to the Author):

In this work, the authors present a latent triethylborane-pyridine complex as a mechanically activated source of radicals. Based on prior work from Pan (co author), triethylborane reduces molecular oxygen to a "radical species" that is capable of initiating polymerization processes.

This is a significant advancement in the field as it allows for the production of polyacrylates (and related materials) under milling conditions. Milling is "green" but also, as the authors demonstrate, provides a means by which to access materials that can't be made using conventional thermal/photochemical approaches (more on that later). I have a few major questions/concerns that need to be addressed before I can recommend for publications:

We thank Reviewer #1 for reviewing our paper and providing their positive feedbacks. We present below a point-by-point response to the comments and questions that were raised. We are confident we addressed the concerns of Reviewer #1 appropriately and improved the manuscript based on their comments.

Comment 1. The authors constantly allude to "radical" species but don't actually define what is doing the initiation. More importantly, from a mechanistic standpoint, how does this species compare to what the active initiating species (singular or plural) is/are in solution (from their ACIE 2018 work)? Does solution-state vs. milling change the nature of these species? I think this type of mechanistic question is important for understanding how this process works and the ability to probe (mass spec, for example) differentiates this work from work that I might find in an applied polymer journal.

Response: Thank you for your insightful comments and for highlighting that we did not properly present the initiation mechanism of our current work in the first version of the manuscript.

In *Angew. Chem. Int. Ed.* **57**, 9430–9433(2018), we demonstrated that the combination of oxygen and Et₃B could initiate and regulate RAFT polymerization. In that publication, the initiation was based on the redox reaction of triethylborane and oxygen molecule, giving rise to the ethyl radical and boryl peroxy radical. The ethyl radical was key initiating radicals.

In the current paper, we designed an organic mechano-labile initiator that could be activated by ball milling to release reactive species required for the oxidative process to produce initiating radicals for

polymerization of vinyl monomers to generate well-defined polymers (Fig. 1). This method of polymerizing vinyl monomers using ball milling not only features open-to-air reaction, low-energy input, operative simplicity, and the avoidance of potentially harmful organic solvents, but also provides us with the unique opportunity to utilize molecular oxygen and applied stress for the controlled polymerization of solid monomers and the bulk synthesis of polymer/perovskite hybrids which are inaccessible by other means.

To verify the radical mechanism involved in the mechanochemical initiation,

Firstly, triethylborane (1M in THF) was easily oxidized in air for one day, giving rise to the oxidized products including Et₂BOEt and (EtO)₂BEt, indicating the high reactivity of Et₃B in air at room temperature.

Secondly, the stability of Et₃B-PyOMe under ambient conditions was studied. The ¹¹B NMR spectroscopy remained unchanged with a clear chemical shift (η) at 0.39 ppm after exposing to air for one week, indicating the stability of the complex in air at room temperature.

Subsequently, we subjected Et₃B-PyOMe to ball milling (20 Hz) under air atmosphere for 60 minutes. The spectra showed that most Et₃B-PyOMe (~97.2%) was transformed into oxidized products including (EtO)₂BEt (~78.5%) and (EtO)₃B (~18.7%). To confirm the radical mechanism for the mechanochemical process, 2,2,6,6-tetramethylpiperidinyl-1-oxide (TEMPO) and 5,5-dimethyl-1-pyrroline n-oxide (DMPO) were respectively used to capture the generated ethyl radicals (**Fig. 3d**). In the ultraviolet visible light (UV-vis) spectra, an absorbance band centered at 469.5 nm for the pristine TEMPO was observed and decreased as ball milling (20 Hz) proceeded. A new absorbance band centered at 630 nm appeared as the TEMPO-trapped intermediates (**Fig. 3e**). Furthermore, the EPR spectra revealed that carbon-centered radicals were generated via oxidative reaction of triethylborane and rapidly captured by DMPO to give stable adducts ($A_N = 15.10$ G, and $A_{H\beta} = 21.50$ G, **Fig. 3f**). Given by this, the mechanism of aerobic mechanochemical activation was outlined in **Fig. 3g**, comprising mechanochemical release of triethylborane and oxidative reaction with oxygen to generate initiating radicals.

To clarify the initiation mechanism of our current work, we made two figures that you will find below and in the revised manuscript.

Figure R1. Illustrative design of aerobic mechano-RDRP inspired by aerobic exercises.

Figure R2. Mechanistic analysis for mechanochemical initiation

To take into account your remark and better represent the initiation process of the aerobic mechano-RDRP over other reported with method, we amended the manuscript as follow (changes in red below, **highlighted in yellow** in the manuscript):

Results (Figure 3):

Organic compounds with labile bonds have been demonstrated to be directly activated by external force including ball milling⁵⁰. To elucidate the mechanochemical activation of $\text{Et}_3\text{B-PyOMe}$, molecular electrostatic potential (MESP) was simulated to monitor the structural evolution of $\text{Et}_3\text{B-PyOMe}$ before and after dissociation. As shown in **Fig.3a**, triethylborane fragment in complex manifested negative electrostatic potentials (blue color), and pyridine fragment exhibited positive electrostatic potentials (red color). After decomplexation, triethylborane was transformed to positive electrostatic potentials (white color) due to the electron-deficient character of boron atom, and nitrogen atom in pyridine ligand performed obvious negative electrostatic potentials (blue color). Fourier transform infrared (FTIR) spectroscopy was further employed to analyze the chemical environment of the pyridine ligand. Along with ball milling, the breathing (at around 1595 cm^{-1}) and

stretching peak (at around 760 cm^{-1}) of the pyridine ring with $\text{Et}_3\text{B-PyOMe}$ was significantly enhanced (**Fig. 3b** and Supplementary Fig. 16)⁵¹, revealing an obvious overlap with that of free PyOMe. Taking Et_3B as the control, there was no signal in the band around 760 cm^{-1} before and after oxidation. These results indicated that the decomplexation of $\text{Et}_3\text{B-PyOMe}$ occurred during ball milling⁵². In addition, ^{11}B NMR spectroscopy was utilized to study the key boron intermediates (**Fig. 3c**). Initially, triethylborane (1M in THF) was easily oxidized in air for one day, giving rise to the oxidized products including Et_2BOEt and $(\text{EtO})_2\text{BEt}$. While in the case of $\text{Et}_3\text{B-PyOMe}$, the ^{11}B NMR spectroscopy remained unchanged with a clear chemical shift (η) at 0.39 ppm after exposing to air for one week, indicating the stability of the complex under ambient conditions. We then subjected $\text{Et}_3\text{B-PyOMe}$ to ball milling (20 Hz) under air atmosphere for 60 minutes. The spectra showed that most $\text{Et}_3\text{B-PyOMe}$ (~97.2%) was transformed into oxidized products including $(\text{EtO})_2\text{BEt}$ (~78.5%) and $(\text{EtO})_3\text{B}$ (~18.7%). To confirm the radical mechanism for the mechanochemical process, 2,2,6,6-tetramethylpiperidinyl-1-oxide (TEMPO) and 5,5-dimethyl-1-pyrroline *n*-oxide (DMPO) were respectively used to capture the generated ethyl radicals (**Fig. 3d**). In the ultraviolet visible light (UV-vis) spectra, an absorbance band centered at 469.5 nm for the pristine TEMPO was observed and decreased as ball milling (20 Hz) proceeded. A new absorbance band centered at 630 nm appeared as the TEMPO-trapped intermediates (**Fig. 3e**)^{24,53}. Furthermore, the EPR spectra revealed that carbon-centered radicals were generated via oxidative reaction of triethylborane and rapidly captured by DMPO to give stable adducts ($A_N = 15.10\text{ G}$, and $A_{\text{HB}} = 21.50\text{ G}$, **Fig. 3f**)^{54,55}. Given by this, the mechanism of aerobic mechanochemical activation was outlined in **Fig. 3g**, comprising mechanochemical release of triethylborane and oxidative reaction with oxygen to generate initiating radicals.

Introduction:

“Reversible-deactivation radical polymerization (RDRP) mediated by the chemical equilibrium between active and dormant species has enabled excellent control over the macromolecular chain structure¹⁸⁻²⁰. Recent advances²¹⁻³³ in mechanochemical radical polymerization have further extended the possibility of RDRP to the mechano-responsive systems including heterogenous curing gels^{31,32}, self-growing polymers^{34,35}, and self-strengthened materials^{36,37}. However, a large amount of metal-based (piezoelectric or triboelectric) materials^{21,22,27,30,32,38} or high-energy input^{24,26} were required to facilitate the formation of initiating radicals for mechano-RDRP, giving rise to the complicated purification procedure for polymer products. Furthermore, as polymerization could be terminated by molecular oxygen, most of these elegant designs were required to be conducted under anaerobic conditions. Recently, enormous effort has been devoted to the removal of dissolved oxygen prior to polymerization, including approaches employing enzymes^{8,9}, microbial metabolisms^{7,10,11}, reducing agents³⁹⁻⁴² or photocatalysts⁴³⁻⁴⁵. The broad success of oxygen-tolerant RDRP hinges on the susceptibility of the chemical conversion of molecular oxygen with high levels of efficiency and selectivity. A much more viable and ambitious solution to this grand challenge would be the development of a metal-free, oxygen-tolerant and low-energy-input mechanochemical system for RDRP.

.”

“During aerobic excises, muscle glycogen particles are broken down under external force, freeing glucose molecules that can be further oxidized through aerobic processes to produce the adenosine triphosphate (ATP) molecules required for biological activities⁴⁵⁻⁴⁷. Inspired by the unique profile of aerobic process, we hypothesized that the regeneration of activators from molecular oxygen could be achieved through a mechanistically distinct approach using mechanical energy.”

Comment 2. In Fig 2d, the authors claim they TEMPO trap a “radical” species. The authors use a solid arrow but don’t provide a yield nor do they provide any experimental evidence of what is formed (aside from UV vis data). You can’t claim to have made a new chemical species (solid arrow) without more rigorous characterization. This point closely relates to my concern above (#1) in that in this paper the authors don’t provide any evidence of how chain growth kicks off..

Response: Thank you for your insightful comments and for highlighting that we did not properly provide enough experimental evidence of “radical” species of our current work in the first version of the manuscript.

To verify the radical species initiating chain growth, 2,2,6,6-tetramethylpiperidiny-1-oxide (TEMPO) and 5,5-dimethyl-1-pyrroline n-oxide (DMPO) were respectively used to capture the generated ethyl radicals (**Fig. 3d**). In the ultraviolet visible light (UV-vis) spectra, an absorbance band centered at 469.5 nm for the pristine TEMPO was observed and decreased as ball milling (20 Hz) proceeded. A new absorbance band centered at 630 nm appeared as the TEMPO-trapped intermediates (**Fig. 3e**). On the basis of Beer-Lambert law ($A/c_0 = \epsilon bc/c_0$), a new species was supposed to be formed (*Macromolecules* **42**, 1067-1078 (2009), *ACS Macro Lett.* **7**, 275-280 (2018)). In addition, EPR spectra were measured to confirm the radical species using DMPO as the radical scavenger. After ball milling, ERP spectra revealed new chemical species as ethyl radical DMPO adduct ($A_N = 15.10$ G, and $A_{H\beta} = 21.50$ G) (*J. Am. Chem. Soc.* **145**, 359-376 (2023), *Nat. Commun.* **14**, 6530 (2023)). As a carbon-centered radical, ethyl radicals have been confirmed to activate chain growth (*Angew. Chem. Int. Ed.* **57**, 9430-9433 (2018), *Macromolecules* **55**, 4056-4063 (2022)).

In summary, we confirmed the initiation process of the aerobic mechano-RDRP based on UV-Vis and EPR. To take into account your remark and better represent the initiation process of the aerobic mechano-RDRP over other reported with method, **we amended the manuscript as follow (changes in red below, highlighted in yellow in the manuscript):**

Results (Figure 3):

*To confirm the radical mechanism for the mechanochemical process, 2,2,6,6-tetramethylpiperidiny-1-oxide (TEMPO) and 5,5-dimethyl-1-pyrroline n-oxide (DMPO) were respectively used to capture the generated ethyl radicals (**Fig. 3d**). In the ultraviolet visible light (UV-vis) spectra, an absorbance band centered at 469.5 nm for the pristine TEMPO was observed and decreased as ball milling (20 Hz) proceeded. A new absorbance band centered at 630 nm appeared as the TEMPO-trapped intermediates (**Fig. 3e**)^{24,53}. Furthermore, the EPR spectra revealed that carbon-centered radicals*

*were generated via oxidative reaction of triethylborane and rapidly captured by DMPO to give stable adducts ($A_N = 15.10$ G, and $A_{HB} = 21.50$ G, **Fig. 3f**)^{54,55}. Given by this, the mechanism of aerobic mechanochemical activation was outlined in **Fig. 3g**, comprising mechanochemical release of triethylborane and oxidative reaction with oxygen to generate initiating radicals.*

Comment 3. I appreciate the polymer/perovskite application but the explanation about what the expected results are is insufficient/confusing. I assume these perovskites emit at 529 nm when well dispersed? But if they cluster they don't emit (photo-RDRP)? And if they sort of disperse but don't crystallize (thermal-RDRP) they also don't emit? This section seems to be not as clear as the rest of the manuscript. I suggest clarifying how all of the controls were designed, what expected results are, and rationale for each piece of data. The cartoons in Fig 4 are even more confusing. Where does the crosslinker come from? The authors haven't made crosslinked materials at all in the paper and then all of a sudden throw it into this experiment. The cartoons make it seem like perovskite and nanocrystal (I am confused about how these are different materials) aren't added into the vessels either. The SI provides a brief prep that suggests that nanocrystals ARE added to the milling jars but unfortunately there is no prep for the control experiments so I can not comment how these were performed. There needs to be some clarity here to help the reader follow what is going on. This section really detracts from an otherwise very nice manuscript.

Response: Thank you for the insightful comments and valuable suggestions. The synthesis of polymer/perovskites hybrids was designed and attempted to show that this mechanochemical approach could be used to synthesize uniform hybrids without solvents. Previous thermal or photo-systems typically adopted polar solvents for the growth and dispersion of perovskites nanocrystals in polymer matrix for efficient and uniform emission. To better convey the advantage of this aerobic mechano-RDRP, we optimized the design for polymer/perovskites hybrids. NIPAM was selected as the functional monomer to coordinate and passivate perovskite nanocrystals (PNCs). As illustrated in **Fig.4a**, PNCs@P(*n*BA-*co*-NIPAM) were synthesized by one-pot ball milling (20 Hz with four 10 mm zirconia balls) without additional solvent and further purification. Well-defined polymers were achieved with high conversion (>80 %, Supplementary Fig.30) and narrow molecular weight distribution ($D = 1.25$, Supplementary Fig.31). The signal intensity analysis of ¹³C NMR spectroscopy (DEPT135, CH₃/CH positive and CH₂ negative) shows that the copolymerization ratio of *n*BA and NIPAM segment is 9.7, which matched well with the feeding ratio (**Fig.4b** and Supplementary Fig.32). The photoluminescence (PL) spectrum of this hybrids was displayed in **Fig. 4c**, with a maximum emissive wavelength around 529 nm and a full width at half-maximum (FWHM) of ~25.2 nm. In addition, the XRD pattern of the hybrid matched the standard card of MAPbBr₃ nanocrystals (**Fig.4d**). "IFE" characters on the glass were made by the injection of PNCs@ P(*n*BA-*co*-NIPAM), affording strong green fluorescence emission under 365 nm ultraviolet irradiation (**Fig.4e**). These results demonstrated the exciting potential of this aerobic mechano-RAFT as an operationally simple, mild and solvent-free route to synthesize polymer/inorganic hybrids from commercial monomers and precursors.

We revised the figure that you will find below and in the revised manuscript.

Figure R3. Polymer/perovskites hybrids by aerobic mechano-RAFT.

Results (Figure 4):

“Unlike previous piezoelectrical^{27,31-33} and triboelectrical⁵⁷ systems depending on a high loading of inorganic powders, this aerobic mechano-RAFT adopted a tiny amount of mechano-labile initiators. Thus, we envisioned this approach could be used for the bottom-up synthesis of polymer/inorganic hybrids without solvent and degassing. The synthesis of polymer/perovskites hybrids was attempted because polymer/perovskites hybrids has received intensive interest on account of the role in light-emitting diodes⁵⁸, photovoltaics⁵⁹, sensors⁶⁰, and thin-film transistors⁶¹. With this in mind, the synthesis of polymer/perovskites hybrids was attempted. nBA was used as the first monomer due to the excellent control of polymerization^{62,63}. NIPAM was selected as the functional monomer to coordinate and passivate perovskite nanocrystals (PNCs)⁶⁴. As illustrated in **Fig.4a**, PNCs@P(nBA-co-NIPAM) were synthesized by one-pot ball milling (20 Hz with four 10 mm zirconia balls) without additional solvent and further purification. Well-defined polymers were achieved with high conversion (>80 %, Supplementary Fig.30) and narrow molecular weight distribution ($\bar{D} = 1.25$, Supplementary Fig.31). The signal intensity analysis of ¹³C NMR spectroscopy (DEPT135, CH₃/CH positive and CH₂ negative) shows that the copolymerization ratio of nBA and NIPAM segment is 9.7, which matched well with the feeding ratio (**Fig.4b** and Supplementary Fig.32). The photoluminescence (PL) spectrum of this hybrids was displayed in **Fig. 4c**, with a maximum emissive wavelength around 529 nm and a full width at half-maximum (FWHM) of ~25.2 nm. In addition, the XRD pattern of the hybrid matched the

standard card of MAPbBr₃ nanocrystals (Fig.4d)⁶⁵. “IFE” characters on the glass were made by the injection of PNCs @ P(nBA-co-NIPAM), affording strong green fluorescence emission under 365 nm ultraviolet irradiation (Fig.4e). These results demonstrated the exciting potential of this aerobic mechano-RAFT as an operationally simple, mild and solvent-free route to synthesize polymer/inorganic hybrids from commercial monomers and precursors.”

Introduction:

“This method of polymerizing vinyl monomers using ball milling not only features open-to-air reaction, low-energy input, operative simplicity, and the avoidance of potentially harmful organic solvents, but also provides us with the unique opportunity to utilize molecular oxygen and applied stress for the controlled polymerization of solid monomers and the bulk synthesis of polymer/perovskite hybrids which are inaccessible by other means.”

Conclusions:

“This method enables the reaction to proceed in air with low-energy input, operative simplicity, and the avoidance of potentially harmful organic solvents. In addition, this approach not only complements the existing mechanochemical approaches for the control of macromolecular structure, but also accesses well-defined polymer/perovskite hybrids without solvent which are inaccessible by other mechanochemical means.”

Comment 4. The IR spectral shifts described in the main text p4 should be qualified with specific structural features.

Response: Thank you for the suggestion. We have assigned the obvious signals in the IR spectra for each compound. The breathing (at around 1595 cm⁻¹) and stretching peak (at around 760 cm⁻¹) of the pyridine ring with Et₃B-PyOMe was significantly enhanced. (Fig.3b and Supplementary Fig. 16), revealing an obvious overlap with that of free PyOMe. Taking Et₃B as the control, there was no signal in the band around 760 cm⁻¹ before and after oxidation. These results indicated that the decomplexation of Et₃B-PyOMe occurred during ball milling.

To clarify the mechanochemical decomplexation of Et₃B-PyOMe during ball milling, we made the figure that you will find below.

Figure R4. FT-IR spectra of Et₃B-PyOMe and related compounds.

Results Figure 3:

“Fourier transform infrared (FTIR) spectroscopy was further employed to analyze the chemical environment of the pyridine ligand. Along with ball milling, the breathing (at around 1595 cm⁻¹) and stretching peak (at around 760 cm⁻¹) of the pyridine ring with Et₃B-PyOMe was significantly enhanced (Fig.3b and Supplementary Fig. 16)⁵⁰, revealing an obvious overlap with that of free PyOMe. Taking Et₃B as the control, there was no signal in the band around 760 cm⁻¹ before and after oxidation. These results indicated that the decomplexation of Et₃B-PyOMe occurred during ball milling⁵¹.”

Comment 5. 2. Do the authors know the conversion to three significant figures as suggested in Fig 3 (and in line text)?

Response: Thank you for the insightful comments and valuable suggestions. We recognize the conversion obtained by ¹H NMR should retain appropriate significant figures.

We revised the figure that you will find below and in the revised manuscript.

Figure R5. Aerobic mechano-RAFT driven by ball-milling

Results (Figure 2):

“To verify the feasibility of this concept, we initially investigated the aerobic mechanochemical reversible addition–fragmentation chain transfer (mechano-RAFT) polymerization as the model method for the polymerization of *n*-butyl acrylate (*n*BA) using as monomer, 2-butylsulfanylthiocarbonylsulfanyl-propionic acid (BTPA) using as the chain transfer agent (CTA), and triethylborane/pyridine complex (*Et*₃B-PyOMe) using as the mechano-labile molecules (Fig. 2a). The pristine transparent liquids switched to transparent and sticky liquids after 2 h ball milling with 30 Hz oscillation frequency (Fig. 2b, up), and the ¹H NMR spectroscopy revealed 81% conversion for the polymerization (Fig. 2c and Supplementary Fig.1). The reactions without ball milling, air or initiator showed little conversion (<5.0 %, Supplementary Fig.2).”

Comment 6. *M_n* values in the main text should have associated units?

Response: Thank you for the suggestion. We would like to point out that *M_n* is dimensionless based on the calibration methods of GPC which is a relative measurement of molecular weight. As we can find in many published papers, *M_n* was recognized as dimensionless. (*J. Am. Chem. Soc.* **117**, 5614-5615 (1995), *J. Am. Chem. Soc.* **145**, 3, 1906-1915 (2023) et al.).

-[R1] Lv, C., He, C. & Pan, X. Oxygen-Initiated and Regulated Controlled Radical Polymerization

under Ambient Conditions. *Angew. Chem. Int. Ed.* **57**, 9430-9433 (2018).

- [R2] Wang, Z. *et al.* Ultrasonication-Induced Aqueous Atom Transfer Radical Polymerization. *ACS Macro Letters* **7**, 275-280 (2018).
- [R3] Lu, C.-H., Huang, C.-F., Kuo, S.-W. & Chang, F.-C. Synthesis and Characterization of Poly(ϵ -caprolactone-*b*-4-vinylpyridine): Initiation, Polymerization, Solution Morphology, and Gold Metalation. *Macromolecules* **42**, 1067-1078 (2009).
- [R4] An, Q. *et al.* Identification of Alkoxy Radicals as Hydrogen Atom Transfer Agents in Ce-Catalyzed C–H Functionalization. *Journal of the American Chemical Society* **145**, 359-376 (2023).
- [R5] Zhong, P.-F. *et al.* Photoelectrochemical oxidative C(sp³)–H borylation of unactivated hydrocarbons. *Nature Communications* **14**, 6530 (2023).
- [R6] Ding, C. *et al.* Piezoelectrically Mediated Reversible Addition–Fragmentation Chain-Transfer Polymerization. *Macromolecules* **55**, 4056-4063 (2022).
- [R6] Hu, H. *et al.* Novel MAPbBr₃ perovskite/ polymer nanocomposites with luminescence and self-healing properties: In suit fabrication and structure characterization. *Optical Materials* **119**, 111405 (2021).
- [R7] Benassi, E. & Fan, H. Quantitative characterisation of the ring normal modes. Pyridine as a study case. *Spectrochimica Acta Part A: Molecular and Biomolecular Spectroscopy* **246**, 119026 (2021).
- [R8] Chakraborty, S. & Dopfer, O. Infrared Spectrum of the Ag⁺–(Pyridine)₂ Ionic Complex: Probing Interactions in Artificial Metal-Mediated Base Pairing. *ChemPhysChem* **12**, 1999-2008 (2011).
- [R9] Wang, J.-S. & Matyjaszewski, K. Controlled/"living" radical polymerization. atom transfer radical polymerization in the presence of transition-metal complexes. *Journal of the American Chemical Society* **117**, 5614-5615 (1995).

Reviewer #2 (Remarks to the Author):

The team described ball-milling RAFT with aerobic initiation. It is clearly stated that ball-milling truly modulated RAFT, but the lack of details blurred the nice achievement. Even it presented new hybrid material fabrication, it is still not enough to convince this development truly expands a realm of synthetic polymer chemistry. Sorry, I can not recommend it for Nature Communications.

We thank reviewer #2 for taking the time to evaluate the present manuscript, and their comment and remarks. We are confident we addressed the concerns of Reviewer #2 appropriately and improved the manuscript based on their comments.

Comment 1. The identity of the initiation process remains unclear. Reference 24 detailed the chemical mechanism of Et₃B and oxygen at room temperature. The authors introduced pyridine to inhibit Et₃B, which is believed to be reactivated during ball-milling. Several spectroscopic pieces of evidence support Et₃B liberation and oxygen activation, but the structure of oxidized Et₃B-Py remains elusive. At the very least, endgroup analysis is necessary to comprehend the initiation process.

Response: Thank you for your kind remind and suggestion. FI-IR and ¹¹B NMR spectra were carefully obtained and comprehensively analyzed to reveal the mechanochemical decomplexation of Et₃B-PyOMe and the structure of oxidized Et₃B-Py. Along with ball milling, the breathing (at around 1595 cm⁻¹) and stretching peak (at around 760 cm⁻¹) of the pyridine ring with Et₃B-PyOMe was significantly enhanced (**Fig.R1a-b**), revealing an obvious overlap with that of free PyOMe. Taking Et₃B as the control, there was no signal in the band around 760 cm⁻¹ before and after oxidation. These results indicated that the decomplexation of Et₃B-PyOMe occurred during ball milling. In addition, ¹¹B NMR spectroscopy was utilized to study the key boron intermediates (**Fig.R1c**). Initially, triethylborane (1M in THF) was easily oxidized in air for one day, giving rise to the oxidized products including Et₂BOEt and (EtO)₂BEt. While in the case of Et₃B-PyOMe, the ¹¹B NMR spectra remained unchanged with a clear chemical shift (η) at 0.39 ppm after exposing to air for one week, indicating the stability of the complex under ambient conditions. We then subjected Et₃B-PyOMe to ball milling (20 Hz) under air atmosphere for 60 minutes. The spectra showed that most Et₃B-PyOMe (~97.2%) was transformed into oxidized products including (EtO)₂BEt (~78.5%) and (EtO)₃B (~18.7%).

End-group analysis was further conducted via MALDI-TOF mass spectroscopy. The strongest peak revealed a molecular weight of 8,054 (DP = 61, **Fig.R2a**) which matched well with the corresponding GPC trace ($M_n = 7,500$, **Fig.R2b**). Another distribution at 8,069 can be attributed to the removal of a methyl group by laser irradiation, and the main peak with intervals of 128.03 corresponded to the molar mass of the *n*BA unit (**Fig.R2a**).

To clarify initiation process during ball milling, we made two figures that you will find below.

Figure R6. Mechanochemical decomplexation of Et₃B-PyOMe a) FT-IR of Et₃B-PyOMe throughout ball milling; b) FT-IR of Et₃B, PyOMe and Et₃B-PyOMe; c) ¹¹B NMR spectra of Et₃B and Et₃B-PyOMe.

Figure R7. PnBA synthesized via the aerobic mechano-RAFT. a) The MALDI-TOF spectrum; b) the GPC trace.

We added the following comments to consider this point:

Results (Figure 3):

“Fourier transform infrared (FTIR) spectroscopy was further employed to analyze the chemical environment of the pyridine ligand. Along with ball milling, the breathing (at around 1595 cm⁻¹) and stretching peak (at around 760 cm⁻¹) of the pyridine ring with Et₃B-PyOMe was significantly enhanced (Fig. 3b and Supplementary Fig. 16)⁵⁰, revealing an obvious overlap with that of free PyOMe. Taking Et₃B as the control, there was no signal in the band around 760 cm⁻¹ before and after oxidation. These results indicated that the decomplexation of Et₃B-PyOMe occurred during ball milling⁵¹. In addition, ¹¹B NMR spectroscopy was utilized to study the key boron intermediates (Fig. 3c). Initially, triethylborane (1M in THF) was easily oxidized in air for one day, giving rise to the oxidized products including Et₂BOEt and (EtO)₂BEt. While in the case of Et₃B-PyOMe, the ¹¹B NMR spectroscopy

remained unchanged with a clear chemical shift (η) at 0.39 ppm after exposing to air for one week, indicating the stability of the complex under ambient conditions. We then subjected Et₃B-PyOMe to ball milling (20 Hz) under air atmosphere for 60 minutes. The spectra showed that most Et₃B-PyOMe (~97.2%) was transformed into oxidized products including (EtO)₂BEt (~78.5%) and (EtO)₃B (~18.7%).”

“Another feature of this aerobic mechano-RAFT is the high retention of chain end for the synthesized polymer. Matrix-assisted laser desorption/ionization-time of flight (MALDI-TOF) mass spectra of PnBA showed that the strongest peak revealed a molecular weight of 8,054 (DP = 61, Supplementary Fig. 13) which matched well with the corresponding GPC trace ($M_n = 7,500$, Supplementary Fig. 14). Another distribution at 8,069 can be attributed to the removal of a methyl group by laser irradiation³³, and the main peak with intervals of 128.03 Da corresponded to the molar mass of the nBA unit (Supplementary Fig. 15).”

Methods:

“**MALDI-TOF mass spectroscopy of PnBA.** PnBA for MALDI-TOF mass spectroscopy were obtained from general aerobic mechano-RAFT procedure (30 Hz, 1h). MALDI-TOF mass spectrometer was from Bruker, Germany. The MALDI instrument was equipped with a 337 nm pulsed nitrogen laser (laser intensity of 50 Hz). The number of laser irradiations was 100 for all mass spectra (delay time of 190 ns), with a 20 kV acceleration voltage. MALDI experiment was carried out using 2,5-DHB as the matrix. The matrix solution was prepared by dissolving 40 mg of 2,5-DHB in 1 mL of THF.”

Comment 2. The use of DFT may not be appropriate. The authors applied directional force to break the coordination between Et₃B and pyridine, which I disagree with. These directional forces might be applicable to chemical bonds embedded in polymer chains. Many studies by Moore, Otsuka, Peterson, Yoon, and others suggest that sufficient chain length is necessary to stretch the chain and break the labile chemical bond in the middle of the chain. When short chains are used, no directional chemical deformation is observed with ball-milling. Therefore, DFT calculations of the small complex, Et₃B:Py, are not properly designed.

Response: Thank you for your kind remind. Ball mill is regarded as non-directional force but has been widely used for the activation of organic reactions. Moreover, organic compounds could be directed activated by external force shown by Otsuka (*Angew. Chem. Int. Ed.* **54**, 6168-6172 (2015)). However, the mechanism lied in the mechanochemical activation of organic compounds are not clear. After judicious consideration, we decide to adjust the description for the DFT since there is no clear and straightforward method to support the mechanochemical activation of small complex.

Alternatively, molecular electrostatic potential (MESP) was simulated to monitor the structural evolution of Et₃B-PyOMe before and after dissociation. As shown in **the figure (see below)**, triethylborane fragment in complex manifested negative electrostatic potentials (blue color), and pyridine fragment exhibited positive electrostatic potentials (red color). After decomplexation, triethylborane was transformed to positive electrostatic potentials (white color) due to the electron-deficient character of boron atom, and nitrogen atom in pyridine ligand performed obvious negative electrostatic potentials (blue color).

Figure R8. MESP analysis of the Et₃B-PyOMe complex

We added the following sentences to the main text (Results)

Results (Figure 3):

“Organic compounds with labile bonds have been demonstrated to be directly activated by external force including ball milling⁴⁹. To elucidate the mechanochemical activation of Et₃B-PyOMe, molecular electrostatic potential (MESP) was simulated to monitor the structural evolution of Et₃B-PyOMe before and after dissociation. As shown in Fig.3a, triethylborane fragment in complex manifested negative electrostatic potentials (blue color), and pyridine fragment exhibited positive electrostatic potentials (red color). After decomplexation, triethylborane was transformed to positive electrostatic potentials (white color) due to the electron-deficient character of boron atom, and nitrogen atom in pyridine ligand performed obvious negative electrostatic potentials (blue color).”

Comment 3. Polymerization without oxygen should be conducted to confirm oxygen's involvement. Additionally, control of chain length by TTC loading is necessary..

Response: Thank you for the suggestion. The polymerization without air was conducted, and a very low conversion (<5%) was obtained, indicating the critical role of oxygen involved in the initiation (**Fig. R4a**). The chain length of PnBA was explored by adjusting the molar ratio of BA: BTPA. We firstly explored the effect of target DP ($DP_T = 100/200/400$). All the reactions achieved high monomer conversion (>70.0%, Supplementary Fig. 11), giving well defined polymers with excellent match between experimental molecular weight and theoretical value as well as low dispersity (**Fig. R4b**).

Furthermore, we tried to synthesize a polymer with $DP_T = 1,000$ and obtained a PnBA with an actual $DP = 950$ and $\bar{D} = 1.29$ through the optimized method (Supplementary Fig. 12, details in **Methods** section), featuring better control targeting the synthesis of high DP polymer compared to the highest DP in previous mechanochemical system (PMMA, $DP = 730$ and $\bar{D} = 1.77$) (*ACS Macro Letters* 12, 26-32 (2023)).

To show the necessary condition of aerobic mechano-RAFT and control of chain length, we made one figure that you will find below.

Controls for aerobic mechano-RAFT

Condition	Conversion
Ball milling 30Hz	81%
Without ball milling	<5%
Without air	<5%
Without initiator	<5%
Under 30°C heating	<5%

Figure R9. Aerobic mechano-RAFT: Control experiments and polymerization targeting various DP.

We added the following sentences to the main text (Results)

Results (Figure 2):

“The pristine transparent liquids switched to transparent and sticky liquids after 2 h ball milling with 30 Hz oscillation frequency (Fig. 2b, up), and the 1H NMR spectroscopy revealed 81% conversion for the polymerization (Fig. 2c and Supplementary Fig.1). The reactions without ball milling, air or initiator showed little conversion (<5.0 %, Supplementary Fig.2). To evaluate the influence of heat generated during ball milling, the temperature inside the milling jar throughout the polymerization was measured by thermography, increasing from 20.9 °C in the beginning to 27.9 °C after reaction (Fig. 2b, down). A control experiment at 30 °C was conducted, no signal of polymer could be observed based on the 1H NMR spectroscopy and GPC trace for the reaction (Fig. 2c and Supplementary Fig.3-4). These results indicated that the polymerization was attributed to mechanochemical activation.”

“To explore whether this low-energy-input approach could be used to synthesize polymers with unprecedented high degree of polymerization (DP) compared to previous mechanochemical approach. We firstly explored the effect of target DP ($DP_T = 100/200/400$). All the reactions achieved high

*monomer conversion (>70.0%, Supplementary Fig. 11), giving well defined polymers with excellent match between experimental molecular weight and theoretical value as well as low dispersity (Fig. 2f). Furthermore, we tried to synthesize a polymer with $DP_T = 1,000$ and obtained a PnBA with an actual $DP = 950$ and $\bar{D} = 1.29$ through the optimized method (Supplementary Fig. 12, details in **Methods** section), featuring better control targeting the synthesis of high DP polymer compared to previous mechanochemical system (PMMA, $DP = 730$ and $\bar{D} = 1.77$)³².*

Comment 4. The scope of the monomers used is narrow. Only liquid monomers were used to achieve homogeneous mixing of monomer, initiator, and CTA, and all the examples are easily accessible through simple liquid polymerization. As mechanochemistry gains popularity, it is essential to explore the types of new polymers that can be synthesized through ball-milling. Previous examples by Bielawski (ATRP), Golder (RDRP), Borchardt (Polyphenylene), and Kim (ROMP) beautifully demonstrated the uniqueness of this approach compared to solution-based methods. While the formation of hybrid materials is commendable, it would be suitable for Nature Communications if you could demonstrate a broader monomer scope.

Response: Thank you for the valuable suggestion. The experiments we conducted in the past months demonstrated that this aerobic mechanochemical approach could be employed for the controlled polymerization of (meth)acrylates, styrenic monomers and solid acrylamides (Identical condition: room temperature, ball milling, minimized dosage of solvents, in air). Essentially, we have optimized corresponding mechanochemical procedure for different monomers based on the physiochemical properties.

The polymerization of (meth)acrylates including *n*-butyl acrylate (*n*BA), ethyl acrylate (EA), tert-butyl acrylate (*t*BA), methyl acrylate (MA) and methyl methacrylate (MMA) was initially attempted under procedure 1 (marked as green at **Table 1, Scheme 1**). Well-defined polymers were achieved for all the acrylates, but the polymerization of MMA revealed a wider molecular weight distribution ($\bar{D} = 1.50$) due to the slow initiating rate of BTPA for methacrylates (Supplementary Fig.17-19). Then, 4-(benzenecarbonothioylsulfanyl)-4-cyanopentanoic acid (BTCPA) was selected as the CTA for the polymerization of MMA with 10 mm balls and an extended reaction time (5 h), afford a higher conversion (~52%, Supplementary Fig.20) and well-defined PMMA ($M_n = 8,300$, $\bar{D} = 1.19$, Supplementary Fig.21). In addition, the polymerization of styrene (St) and vinyl naphthalene (VN)) and vinyl carbazole (VC) was also performed (Blue region in **Table 1**, procedure 2 of **Scheme 1**). All the reactions produced well-defined polymers with predetermined molecular weight and narrow molecular weight distribution (Supplementary Fig.22-25), indicating that the aerobic mechano-RAFT could be employed for the polymerization of low-activity and solid monomers at room temperature. We further tried to polymerize solid acrylamide monomers using this mechanochemical approach to circumvent the phase transition of polyacrylamide during thermal polymerization. Procedure 3, *S,S'*-bis(*R,R'*-dimethyl-*R''*-acetic acid)-trithiocarbonate (ATTC) as the CTA coupled with stainless steel jar and balls, was performed for the polymerization of solid acrylamide monomers. The polymerization for *N*-isopropyl acrylamide (NIPAM) and *N*-phenylacrylamide (NPA) monomers achieved nearly complete conversion (no monomer signal, Supplementary Fig.26-27) and narrow molecular weight distribution (marked as pink at **Table 1**, Supplementary Fig.28) after 5 h (30 Hz with 4*10 mm

stainless steel balls, procedure 3 of **Scheme 1**). It's essential to explore the upper molecular-weight limit of this approach as ball milling with stainless steel jar will generate strong mechanical force, giving rise to mechanochemical degradation of polymers. The GPC traces revealed a single peak with a molecular-weight range of 1.14~1.52 with increasing the molar ratio of monomer to CTA from 50 to 200 (Supplementary Fig.29). While in the case of $DP_T = 500$, a bimodal peak was observed in the GPC traces, indicating that the polymerization was out of control due to the high-energy mechanical input and highly viscous reaction.

To better convey the versatility of this aerobic mechanochemical approach, we revised the figure and added a table as below.

Scheme R1. Procedures for aerobic mechano-RAFT polymerization regarding various monomers.

Table R1 Monomer scope for aerobic mechano-RAFT

Entry	Monomer	DP_T	Conv. (%) ^a	$M_{n,th}$ ^b	$M_{n,GPC}$ ^c	\mathcal{D} ^c	I^* (%) ^d
1	n BA	200	81	21000	20800	1.11	>99
2	EA	200	75	15200	13400	1.14	>99
3	t BA	200	65	16900	12600	1.16	>99
4	MA	200	65	11400	15100	1.09	76
5	MMA	200	32	6600	8900	1.50	74
6	St	200	32	7100	6100	1.22	>99
7	VN	100	30	4800	4400	1.21	>99
8	VC	100	25	5050	5400	1.45	>94
9	NIPAM	100	>95	11600	25800	1.36	45
10	NPA	100	>95	15000	18900	1.19	79

^a Conversion was determined by ¹H NMR spectroscopy. ^b $M_{n,theo} = M_{end\ group} + [M]_0/[CTA]_0 \times conversion \times M_{monomer}$. ^c M_n and M_w/M_n were determined by GPC. ^d Initiator efficiency (I^*) = $M_{n,theo}/M_{n,exp} \times 100$.

We revised the following comment in the manuscript to take into account this observation, and the above figure and added as Scheme 1 and Table 1.

*“The monomer scope of this approach was extended to a variety of vinyl monomers including solid monomers as depicted in **Table 1**. The polymerization of (meth)acrylates including *n*-butyl acrylate (*n*BA), ethyl acrylate (EA), *tert*-butyl acrylate (*t*BA), methyl acrylate (MA) and methyl methacrylate (MMA) was initially attempted under procedure 1 (marked as green at **Table 1**, **Scheme 1**). Well-defined polymers were achieved for all the acrylates, but the polymerization of MMA revealed a wider molecular weight distribution ($\mathcal{D} = 1.50$) due to the slow initiating rate of BTPA for methacrylates (Supplementary Fig.17-19). Then, 4-(benzenecarbonothioylsulfanyl)-4-cyanopentanoic acid (BTCPA)*

was selected as the CTA for the polymerization of MMA with 10 mm balls and an extended reaction time (5 h), afford a higher conversion (~52%, Supplementary Fig.20) and well-defined PMMA ($M_n = 8,300$, $\mathcal{D} = 1.19$, Supplementary Fig.21). In addition, the polymerization of styrene (St) and vinyl naphthalene (VN) and vinyl carbazole (VC) was also performed (Blue region in **Table 1**, procedure 2 of **Scheme 1**). All the reactions produced well-defined polymers with predetermined molecular weight and narrow molecular weight distribution (Supplementary Fig.22-25), indicating that the aerobic mechano-RAFT could be employed for the polymerization of low-activity and solid monomers at room temperature. We further tried to polymerize solid acrylamide monomers using this mechanochemical approach to circumvent the phase transition of polyacrylamide during thermal polymerization^{55,56}. Procedure 3, *S,S'*-bis(*R,R'*-dimethyl-*R''*-acetic acid)-trithiocarbonate (ATTC) as the CTA coupled with stainless steel jar and balls, was performed for the polymerization of solid acrylamide monomers. The polymerization for *N*-isopropyl acrylamide (NIPAM) and *N*-phenylacrylamide (NPA) monomers achieved nearly complete conversion (no monomer signal, Supplementary Fig.26-27) and narrow molecular weight distribution (marked as pink at **Table 1**, Supplementary Fig.28) after 5 h (30 Hz with 4*10 mm stainless steel balls, procedure 3 of **Scheme 1**). It's essential to explore the upper molecular-weight limit of this approach as ball milling with stainless steel jar will generate strong mechanical force, giving rise to mechanochemical degradation of polymers. The GPC traces revealed a single peak with a molecular-weight range of 1.14~1.52 with increasing the molar ratio of monomer to CTA from 50 to 200 (Supplementary Fig.29). While in the case of $DP_T = 500$, a bimodal peak was observed in the GPC traces, indicating that the polymerization was out of control due to the high-energy mechanical input and highly viscous reaction.”

Comment 5. It is a well-known fact that ball-milling can lead to the degradation of high molecular-weight polymers. All the examples in Table 1 showed products with low molecular weights, which may be stable under ball-milling conditions. It would be beneficial to pursue the synthesis of high molecular-weight products, preferably exceeding 50 kDa.

Response: Thank you for this comment. As shown above, in our current work, the polymerization of *n*BA could access a $M_n = 121800$ and $\mathcal{D} = 1.29$ through the aerobic mechanochemical approach (Supplementary Fig. 12, details in **Methods** section).

Figure R10. PnBA targeting various DPn via this aerobic mechanochemical approach

We added the details in the *Methods* section.

“In 35 mL zirconium oxide jar equipped with four 8 mm zirconium oxide balls, 7.2 mL (50.4 mM, 1000 equiv.) butyl acrylate monomer, 0.012 g (0.050 mM, 1 equiv.) BTPA, 0.016 g (0.076 mM, 3 equiv.) Et₃B-PyOMe and 200 μ L DMF (0.03 mL/g) were mixed. The final reaction mixtures were ball-milled at 30 Hz for 2 h.”

-[R1] Imato, K. *et al.* Mechanophores with a Reversible Radical System and Freezing-Induced Mechanochemistry in Polymer Solutions and Gels. *Angewandte Chemie International Edition* **54**, 6168-6172 (2015).

Reviewer #3 (Remarks to the Author):

Feng et al. present in this manuscript a novel approach for conducting reversible addition-fragmentation chain transfer (RAFT) polymerization. Rather than utilizing a traditional thermal radical initiator, the authors utilized a triethylborane/pyridine (Et₃B-Py) complex and molecular oxygen as a radical source. Specifically, Et₃B released from the Et₃B-Py complex reacts with oxygen to generate radicals. This manuscript brings to my mind a paper by Fedorov and colleagues from over a decade ago, in which a similar alkylborane complex was used to initiate free radical polymerization (*Macromolecules* 2007, 40, 10, 3554-3560). Additionally, the Pan group has also leveraged the reaction between triethylborane and oxygen to facilitate similar RAFT polymerization (*Angew. Chem.* 2018, 130, 9574-9577). Overall, the use of alkylboranes in conventional and controlled radical polymerization is well-documented in the literature (see this review: *J. Polym. Sci.* 2020, 58, 14 - 19). Therefore, the novelty and significance of this work in comparison to existing literature are limited.

I acknowledge that in this study, ball milling was employed to release Et₃B from the complex, whereas Fedorov and colleagues used light. Furthermore, the authors also applied this new approach to synthesize a polymer/perovskite hybrid material with luminescent properties. However, I would argue that such hybrid materials can be readily obtained through similar RAFT polymerization methods (*ACS Energy Lett.* 2022, 7, 2, 610 - 616) or numerous other techniques (refer to the excellent review: *Adv. Mater.* 2021, 33, 2005888). This work presents an alternative approach rather than the sole method to synthesize the hybrid materials.

Altogether, while this work introduces a new approach to RAFT polymerization, its novelty and significance are relatively low. Therefore, I recommend publishing this work in a specialized polymer journal. The following recommendations aim to rectify flaws in data interpretation and enhance the quality of the manuscript.

We thank Reviewer #3 for reviewing our paper and providing their positive feedback. We present below a point-by-point response to the comments and questions that were raised. We are confident we addressed the concerns of Reviewer #3 appropriately and improved the manuscript based on their comments.

Response: Thank you for your insightful comments and for highlighting that we did not properly present the novelty of our current work in the first version of the manuscript. We welcome the opportunity to shed light on different aspects and contributions of this research in order to help highlight how the present manuscript represents a conceptual advance; we then corrected the manuscript to reflect this process better.

Mechanochemistry, which was named as one of 10 chemistry innovations by IUPAC that will change the world in 2019, is undergoing an exciting period of rediscovery and renaissance, as they can generate materials along reaction pathways that are inaccessible through thermal, electrical or light-activated processes. These works either have major breakthroughs in principle or have new applications. The importance and novelty of these works may vary, but they all have unique meaning. Indeed, any progress in chemistry, principles or applications related to mechanochemical radical polymerization is a contribution to the field. However, it doesn't mean that the research in this field

are far from saturated as mechanochemistry is being recognized as a parallel chemical approach with thermochemistry, photochemistry and electrochemistry (such as *Science* **366**,1500-1504(2019), *Science* **380**,1248-1252(2023), *Science* **381**,302-306(2023), *Science* **380**,1053-1058(2023), *Nat. Rev. Chem.* **7**, 51–65 (2023), *Nat. Chem.* 2024. Doi: 10.1038/s41557-024-01508-x).

First, we would like to explicitly mention the novelty of the current work does not lie in the reporting of mechanochemical radical polymerization itself. The first publication of photocontrolled radical polymerization was reported by Craig and colleagues by the utilization of an iridium-based complex as photoredox catalyst (*Angew. Chem. Int. Ed.*, **51**, 8850-8853 (2012)). Later, enormous efforts were devoted to the discovery of metal-free photocatalyst for CRP driven by photoirradiation from UV to visible light (*J. Am. Chem. Soc.* **136**, 16096–16101(2014), *Science* **352**,1082-1086(2016)). In the case of mechanochemically controlled radical polymerization, Aaron et al., for the first time, demonstrated that ultrasonic agitation in presence of metal-based piezoelectric materials (BaTiO₃) generates electronic potential that is sufficient for the reduction of Cu^{II}/Me₆TREN into Cu^I/Me₆TREN to started ATRP (*Nat. Chem.* **9**, 135-139 (2017)). Subsequently, a number of papers were reported to lower the frequency or intensity of mechanical energy (*Nat. Mater.* **20**, 869–874 (2021), *Angew. Chem. Int. Ed.* **62**, e202215733 (2023)) to reduce side reactions by the strong mechanical force. However, a large amount of metal-based (piezoelectric or triboelectric) materials or high-energy input were required to facilitate the formation of initiating radicals for mechano-RDRP, giving rise to the complicated purification procedure for polymer products. Furthermore, as polymerization could be terminated by molecular oxygen, most of these elegant designs were required to be conducted under anaerobic conditions. A much more viable and ambitious solution to this grand challenge would be the development of a metal-free, oxygen-tolerant and low-energy-input mechanochemical system for RDRP.

In the current paper, we employed an organic mechano-labile initiator that could be activated by ball milling to release reactive species required for the oxidative process to produce initiating radicals for in-air polymerization of vinyl monomers to generate well-defined polymers based on (meth)acrylates, styrenic monomers and solid acrylamides as well as polymer/perovskite hybrids without solvent at room temperature which are inaccessible by other means. This strategy has not been previously reported.

Apart from the demonstration of a reaction that has not been previously reported for mechanochemical radical polymerization (controlled radical polymerization by piezoelectricity, triboelectricity or sonochemistry under inert atmosphere), we show that:

1. The organo-driven aerobic mechano-RDRP could be conducted in air through the sole mechanochemical decomplexation of triethylborane/pyridine (Et₃B-Py) complex that is induced by ball-milling. To the best of our knowledge, the mechano-RDRP based on the combinative attributes from molecular oxygen and mechanical forces has not yet been reported in the literature.
2. We provide evidence the ability of this organo-driven aerobic mechano-RDRP could be used to polymerize a wide array of monomers including (meth)acrylates, styrenic monomers and

solid acrylamides at room temperature without additional solvents. This could hardly be achieved by the thermal or photo-initiation system. This demonstration cements previous hypotheses proposed and the unique advantage of mechanochemistry over photochemistry or thermochemistry.

3. This organo-driven aerobic mechano-RDRP expands the range of methods available for researchers to fabricate uniform and high-performance polymer/inorganic hybrids via the bottom-up strategy.

To clarify the difference between previous papers and the present one, we made an illustration that you will find below.

Figure R11. Illustration of the difference between this work and pervious works.

Secondly, we want to mention that organo-driven aerobic mechano-RDRP does hold significant advantages over the other published papers.

1. The triethylborane/pyridine ($\text{Et}_3\text{B-Py}$) complex is an organic compound, with a simple design. Despite that, the process exceeding the performances of most RDRP experiments for in terms of polymerization rate, controllability, and monomer scope (See Table 1).
2. This approach is a form of oxygen-tolerant reaction system capable of conducting in air. The reported methods using ultrasound or ball milling based on high loadings of piezoelectric materials we found were all conducted under inert atmosphere. Open-to-air polymerization is of interest for practical applications, as this will simplify the equipment for scale-up synthesis.
3. This mechanochemical approach is a solventless method to access polymers and

polymer/inorganic hybrids, which complements the photochemical, electrochemical and thermochemical approaches.

4. This mechanochemical approach could be used to access pure polymers without complicated purification procedures, and polymer/inorganic hybrids.
5. This approach is a solely organo-driven system that does not require the use of inorganic powders or metal catalysts, but is rather based on widely available commercial organic compounds.

Thirdly, the synthesis of polymer/perovskites hybrids was designed and attempted to show that this mechanochemical approach could be used to synthesize uniform hybrids without solvents. We would like to bring to your attention that previous method including the listed references (ACS Energy Lett. 2022, 7, 2, 610–616, Adv. Mater. 2021, 33, 2005888) typically adopted a large amount of polar solvents for the growth and dispersion of perovskites nanocrystals in polymer matrix for efficient and uniform emission.

In our current work, no additional solvents were used throughout the mechanochemical synthesis of polymer/perovskites hybrids. To better convey the advantage of this aerobic mechano-RDRP, we optimized the design for polymer/perovskites hybrids. NIPAM was selected as the functional monomer to coordinate and passivate perovskite nanocrystals (PNCs). As illustrated in **Fig.4a**, PNCs@P(*n*BA-*co*-NIPAM) were synthesized by one-pot ball milling (20 Hz with four 10 mm zirconia balls) without additional solvent and further purification. Well-defined polymers were achieved with high conversion (>80 %, Supplementary Fig.30) and narrow molecular weight distribution ($D = 1.25$, Supplementary Fig.31). The signal intensity analysis of ^{13}C NMR spectroscopy (DEPT135, CH_3/CH positive and CH_2 negative) shows that the copolymerization ratio of *n*BA and NIPAM segment is 9.7, which matched well with the feeding ratio (**Fig.4b** and Supplementary Fig.32). The photoluminescence (PL) spectrum of this hybrids was displayed in **Fig. 4c**, with a maximum emissive wavelength around 529 nm and a full width at half-maximum (FWHM) of ~ 25.2 nm. In addition, the XRD pattern of the hybrid matched the standard card of MAPbBr₃ nanocrystals (**Fig.4d**). “IFE” characters on the glass were made by the injection of PNCs @ P(*n*BA-*co*-NIPAM), affording strong green fluorescence emission under 365 nm ultraviolet irradiation (**Fig.4e**). These results demonstrated the exciting potential of this aerobic mechano-RAFT as an operationally simple, mild and solvent-free route to synthesize polymer/inorganic hybrids from commercial monomers and precursors.

Results (Figure 4):

Figure 12. Polymer/perovskites hybrids produced by aerobic mechano-RAFT.

There are the obvious advantages of aerobic mechano-RDRP over many methods currently developed. The mere fact that the mechanochemical RDRP process can be tuned by purely organic compounds is, in our humble opinion a significant advance of the aerobic mechano-RDRP.

To take into account your remark and convey better the advancement that represent the current work and the advantages of aerobic mechano-RDRP over other reported with method, we amended the manuscript as follow (changes in red below, highlighted in yellow in the manuscript):

Conclusion:

"In summary, molecular oxygen and mechanical forces have been synergistically utilized to conduct aerobic mechanochemical reversible-deactivation radical polymerization by the deliberate design of an Et₃B/pyridine complex as the mechano-labile initiator that could release free Et₃B upon ball milling. The released Et₃B could further react with molecular oxygen to generate active radicals to induce solventless RAFT polymerization of a wide array of monomers including (meth)acrylates, styrenic monomers and solid acrylamides at room temperature, with excellent control over chain length, dispersity and high chain-end fidelity. This method enables the reaction to proceed in air with low-energy input, operative simplicity, and the avoidance of potentially harmful organic solvents. In addition, this approach not only complements the existing mechanochemical approaches for the control of macromolecular structure, but also accesses well-defined polymer/perovskite hybrids

without solvent which are inaccessible by other mechanochemical means.”

Introduction:

“Reversible-deactivation radical polymerization (RDRP) mediated by the chemical equilibrium between active and dormant species has enabled excellent control over the macromolecular chain structure¹⁸⁻²⁰. Recent advances²¹⁻³³ in mechanochemical radical polymerization have further extended the possibility of RDRP to the mechano-responsive systems including heterogenous curing gels^{31,32}, self-growing polymers^{34,35}, and self-strengthened materials^{36,37}. However, a large amount of metal-based (piezoelectric or triboelectric) materials^{21,22,27,30,32,38} or high-energy input^{24,26} were required to facilitate the formation of initiating radicals for mechano-RDRP, giving rise to the complicated purification procedure for polymer products. Furthermore, as polymerization could be terminated by molecular oxygen, most of these elegant designs were required to be conducted under anaerobic conditions. Recently, enormous effort has been devoted to the removal of dissolved oxygen prior to polymerization, including approaches employing enzymes^{8,9}, microbial metabolisms^{7,10,11}, reducing agents³⁹⁻⁴² or photocatalysts⁴³⁻⁴⁵. The broad success of oxygen-tolerant RDRP hinges on the susceptibility of the chemical conversion of molecular oxygen with high levels of efficiency and selectivity. A much more viable and ambitious solution to this grand challenge would be the development of a metal-free, oxygen-tolerant and low-energy-input mechanochemical system for RDRP.

*During aerobic excises, muscle glycogen particles are broken down under external force, freeing glucose molecules that can be further oxidized through aerobic processes to produce the adenosine triphosphate (ATP) molecules required for biological activities⁴⁵⁻⁴⁷. Inspired by the unique profile of aerobic process, we hypothesized that the regeneration of activators from molecular oxygen could be achieved through a mechanistically distinct approach using mechanical energy. Herein, we designed an organic mechano-labile initiator that could be activated by ball milling to release reactive species required for the oxidative process to produce initiating radicals for polymerization of vinyl monomers to generate well-defined polymers (**Fig. 1**). This method of polymerizing vinyl monomers using ball milling not only features open-to-air reaction, low-energy input, operative simplicity, and the avoidance of potentially harmful organic solvents, but also provides us with the unique opportunity to utilize molecular oxygen and applied stress for the controlled polymerization of solid monomers and the bulk synthesis of polymer/perovskite hybrids which are inaccessible by other means.*

Results (Figure 4):

“Unlike previous piezoelectrical^{27,31-33} and triboelectrical⁵⁷ systems depending on a high loading of inorganic powders, this aerobic mechano-RAFT adopted a tiny amount of mechano-labile initiators. Thus, we envisioned this approach could be used for the bottom-up synthesis of polymer/inorganic hybrids without solvent and degassing. The synthesis of polymer/perovskites hybrids was attempted because polymer/perovskites hybrids has received intensive interest on account of the role in light-emitting diodes⁵⁸, photovoltaics⁵⁹, sensors⁶⁰, and thin-film transistors⁶¹. With this in mind, the synthesis of polymer/perovskites hybrids was attempted. nBA was used as the first monomer due to the excellent control of polymerization^{62,63}. NIPAM was selected as the functional monomer to coordinate and

passivate perovskite nanocrystals (PNCs)⁶⁴. As illustrated in Fig.4a, PNCs@P(nBA-co-NIPAM) were synthesized by one-pot ball milling (20 Hz with four 10 mm zirconia balls) without additional solvent and further purification. Well-defined polymers were achieved with high conversion (>80 %, Supplementary Fig.30) and narrow molecular weight distribution ($\mathcal{D} = 1.25$, Supplementary Fig.31). The signal intensity analysis of ¹³C NMR spectroscopy (DEPT135, CH₃/CH positive and CH₂ negative) shows that the copolymerization ratio of nBA and NIPAM segment is 9.7, which matched well with the feeding ratio (Fig.4b and Supplementary Fig.32). The photoluminescence (PL) spectrum of this hybrids was displayed in Fig. 4c, with a maximum emissive wavelength around 529 nm and a full width at half-maximum (FWHM) of ~25.2 nm. In addition, the XRD pattern of the hybrid matched the standard card of MAPbBr₃ nanocrystals (Fig.4d)⁶⁵. “IFE” characters on the glass were made by the injection of PNCs @ P(nBA-co-NIPAM), affording strong green fluorescence emission under 365 nm ultraviolet irradiation (Fig.4e). These results demonstrated the exciting potential of this aerobic mechano-RAFT as an operationally simple, mild and solvent-free route to synthesize polymer/inorganic hybrids from commercial monomers and precursors.”

Comment 1. This approach is based on RAFT polymerization and therefore, the absence of any mention of RAFT polymerization in the entire manuscript is a significant oversight. I strongly recommend that the authors add a paragraph about RAFT polymerization in the introduction and discuss this polymerization technique throughout the paper.

Response: Thank you for your insightful comments. We would like to point out that this aerobic mechanochemical approach was proposed and demonstrated to address the challenge of previous mechano-RDRP requiring a high loading of inorganic materials, high-energy input and inert atmosphere. As a proof-of-concept, RAFT as a model polymerization method was successfully attempted. This initiation system could be potentially extended to other CRP techniques such as ATRP, iodine-transfer polymerization and telluride mediated polymerization. We are working on these systems and will present them in more details in the future work.

To clarify this, we revised the sentences as below.

Results:

“The generation of activating adenosine triphosphate (ATP) throughout aerobic exercise relies on the coupling of mechanochemical activation of glycogen and oxidation of glucose⁴⁸. Inspired by this, we designed a triethylborane/pyridine (Et₃B-Py) complex as the latent initiator consisting of triethylborane and an electron-donating pyridine for the mechanochemical conversion of oxygen into activators to enable RDRP. This design arose from our previous findings that triethylborane could react with molecular oxygen in air to form initiating radicals for polymerization³⁸. The reactivity of triethylborane was expected to be blocked by the judicious selection of ligand, then restored in response to ball milling (Fig. 2a). To verify the feasibility of this concept, we initially investigated the aerobic mechanochemical reversible addition–fragmentation chain transfer (mechano-RAFT) polymerization as the model method for the polymerization of n-butyl acrylate (nBA) using as monomer, 2-butylsulfanyl-thiocarbonylsulfanyl-propionic acid (BTPA) using as the chain transfer agent (CTA), and triethylborane/pyridine complex (Et₃B-PyOMe) using as the mechano-labile molecules (Fig. 2a).”

Introduction:

“As polymerization could be terminated by molecular oxygen, most of these elegant designs were required to be conducted under anaerobic conditions. In addition, a large amount of inorganic (piezoelectric or triboelectric) materials or high-energy input were required to facilitate the formation of initiating radicals for mechano-RDRP, giving rise to the complicated purification procedure for polymer products.”

Comment 2. The term Reversible-Deactivation Radical Polymerization (RDRP) includes RAFT, ATRP, NMP, etc., so it is overly general to use in the title, abstract, and main text. Does this approach also apply to ATRP and NMP?

Response: Thank you for the insightful comments. This approach could be applied to ATRP, and we will present them in more details in the future work.

Comment 3. On page 4, the authors conclude that "These results suggested that ball milling could change the binding state of pyridine ligands in the Et₃B-Py complex and induce the oxidation of Et₃B." This statement appears to be incorrect. Does the ball milling or molecular oxygen induce the oxidation of Et₃B?

Response: Thank you for the valuable suggestions. The experiments we conducted suggested that ball milling could induced the decomplexation of Et₃B-PyOMe complex to release free Et₃B for the further oxidative reactions to generate initiating radicals. FI-IR and ¹¹B NMR spectra were carefully obtained and comprehensively analyzed to reveal the mechanochemical decomplexation of Et₃B-PyOMe and the structure of oxidized Et₃B-Py. Along with ball milling, the breathing (at around 1595 cm⁻¹) and stretching peak (at around 760 cm⁻¹) of the pyridine ring with Et₃B-PyOMe was significantly enhanced (**Fig. R1a-b**), revealing an obvious overlap with that of free PyOMe. Taking Et₃B as the control, there was no signal in the band around 760 cm⁻¹ before and after oxidation. These results indicated that the decomplexation of Et₃B-PyOMe occurred during ball milling. In addition, ¹¹B NMR spectroscopy was utilized to study the key boron intermediates (**Fig. R1c**). Initially, triethylborane (1M in THF) was easily oxidized in air for one day, giving rise to the oxidized products including Et₂BOEt and (EtO)₂BEt. While in the case of Et₃B-PyOMe, the ¹¹B NMR spectra remained unchanged with a clear chemical shift (η) at 0.39 ppm after exposing to air for one week, indicating the stability of the complex under ambient conditions. We then subjected Et₃B-PyOMe to ball milling (20 Hz) under air atmosphere for 60 minutes. The spectra showed that most Et₃B-PyOMe (~97.2%) was transformed into oxidized products including (EtO)₂BEt (~78.5%) and (EtO)₃B (~18.7%).

To clarify the mechanochemical decomplexation of Et₃B-PyOMe during ball milling, we made a figure that you will find below.

Figure R13. Mechanochemical decomplexation of Et₃B-PyOMe a) FT-IR of Et₃B-PyOMe throughout ball milling; b) FT-IR of Et₃B, PyOMe and Et₃B-PyOMe; c) ¹¹B NMR spectra of Et₃B and Et₃B-PyOMe.

We added the following comments to consider this point:

Results (Figure 3):

“Fourier transform infrared (FTIR) spectroscopy was further employed to analyze the chemical environment of the pyridine ligand. Along with ball milling, the breathing (at around 1595 cm⁻¹) and stretching peak (at around 760 cm⁻¹) of the pyridine ring with Et₃B-PyOMe was significantly enhanced (Fig. 3b and Supplementary Fig. 16)⁵⁰, revealing an obvious overlap with that of free PyOMe. Taking Et₃B as the control, there was no signal in the band around 760 cm⁻¹ before and after oxidation. These results indicated that the decomplexation of Et₃B-PyOMe occurred during ball milling⁵¹. In addition, ¹¹B NMR spectroscopy was utilized to study the key boron intermediates (Fig. 3c). Initially, triethylborane (1M in THF) was easily oxidized in air for one day, giving rise to the oxidized products including Et₂BOEt and (EtO)₂BEt. While in the case of Et₃B-PyOMe, the ¹¹B NMR spectroscopy remained unchanged with a clear chemical shift (η) at 0.39 ppm after exposing to air for one week, indicating the stability of the complex under ambient conditions. We then subjected Et₃B-PyOMe to ball milling (20 Hz) under air atmosphere for 60 minutes. The spectra showed that most Et₃B-PyOMe (~97.2%) was transformed into oxidized products including (EtO)₂BEt (~78.5%) and (EtO)₃B (~18.7%).”

Comment 4. It is widely recognized that ball milling can disrupt bulk complex structures, increase surface area, and enhance reactivity. However, there is no scientific evidence to support the claim that ball milling can induce stretching and bond dissociation as depicted in Figure 2e. How can ball milling simulate external force at two specific locations as shown in Figure 2e? This DFT simulation is not convincing evidence to explain the bond dissociation and release of Et₃B.?

Response: Thank you for your insightful comments. Ball mill is regarded as non-directional force but has been widely used for the activation of organic reactions. Moreover, organic compounds could be directed activated by external force shown by Otsuka (*Angew. Chem. Int. Ed.* **54**, 6168-6172 (2015)).

However, the mechanism lied in the mechanochemical activation of organic compounds are not clear. After judicious consideration, we decide to adjust the description for the DFT.

Alternatively, molecular electrostatic potential (MESP) was simulated to monitor the structural evolution of Et₃B-PyOMe before and after dissociation. As shown in **the figure (see below)**, triethylborane fragment in complex manifested negative electrostatic potentials (blue color), and pyridine fragment exhibited positive electrostatic potentials (red color). After decomplexation, triethylborane was transformed to positive electrostatic potentials (white color) due to the electron-deficient character of boron atom, and nitrogen atom in pyridine ligand performed obvious negative electrostatic potentials (blue color).

Figure R14. MESP analysis of the Et₃B-PyOMe complex

We added the following sentences to the main text (Results)

Results (Figure 3):

“Organic compounds with labile bonds have been demonstrated to be directly activated by external force including ball milling⁴⁹. To elucidate the mechanochemical activation of Et₃B-PyOMe, molecular electrostatic potential (MESP) was simulated to monitor the structural evolution of Et₃B-PyOMe before and after dissociation. As shown in Fig.3a, triethylborane fragment in complex manifested negative electrostatic potentials (blue color), and pyridine fragment exhibited positive electrostatic potentials (red color). After decomplexation, triethylborane was transformed to positive electrostatic potentials (white color) due to the electron-deficient character of boron atom, and nitrogen atom in pyridine ligand performed obvious negative electrostatic potentials (blue color).”

Comment 5. In Figure 2b, I recommend adding a control ¹¹B NMR spectroscopy of oxidized product without using ball milling (Et₃B reacts with oxygen).

Response: Thanks for the valuable suggestion. This has been addressed in the response to **Comment 3**.

Comment 6. The comparison of aerobic mechano-RDRP in this work and skeletal muscle in aerobic exercise in Figure 1 and the introduction is not scientifically relevant. Aerobic physical activities can positively impact tissue regeneration through the release of growth factors, improvement of blood flow, and overall enhancement of tissue microenvironments. During aerobic exercise, the body supplies energy to skeletal muscles through various pathways, including those that do not always require oxygen such as Glycolysis. Additionally, the force applied during aerobic exercise does not aid in energy production by consuming oxygen, as in this chemistry approach, where the force from ball milling helps release Et₃B and produce radicals. Such a comparison makes the work sound like a nature-mimicking system, but it is not accurate. I suggest removing this comparison.

Response: Thanks for your insightful comments. During aerobic excises, muscle glycogen particles are broken down under external force, freeing glucose molecules that can be further oxidized through aerobic processes to produce the adenosine triphosphate (ATP) molecules required for biological activities (*Acta Physiologica Scandinavica* **71**, 129-139 (1967), **125**, 395-405 (1985), *Nutrition Reviews* **76**, 243-259 (2018)). Inspired by the unique profile of aerobic process, we hypothesized that the regeneration of activators from molecular oxygen could be achieved through a mechanistically distinct approach using mechanical energy. Herein, we designed an organic mechano-labile initiator that could be activated by ball milling to release reactive species required for the oxidative process to produce initiating radicals for polymerization of vinyl monomers to generate well-defined polymers.

To clarify our strategy biomimicking aerobic exercises, we made a figure you can see below.

Figure R15. Illustrative design of aerobic mechano-RDRP inspired by aerobic exercises.

We added the following sentences to the main text (Results)

Introduction:

“During aerobic excises, muscle glycogen particles are broken down under external force, freeing glucose molecules that can be further oxidized through aerobic processes to produce the adenosine triphosphate (ATP) molecules required for biological activities⁴⁵⁻⁴⁷. Inspired by the unique profile of

aerobic process, we hypothesized that the regeneration of activators from molecular oxygen could be achieved through a mechanistically distinct approach using mechanical energy. Herein, we designed an organic mechano-labile initiator that could be activated by ball milling to release reactive species required for the oxidative process to produce initiating radicals for polymerization of vinyl monomers to generate well-defined polymers (Fig. 1).”

Comment 7. The chain extension data in Figure S24 shows very little shift (from 26000 to 30000), which is not convincing evidence for high end group fidelity. The authors should aim to achieve a larger increase in molecular weight.

Response: Thanks for the valuable suggestion. Chain extension was re-conducted to examine the chain-end fidelity of the synthesized polymer by this approach. *n*BA was polymerized under identical conditions to give first block *Pn*BA ($M_n = 14,900$ and $D = 1.15$). Ethyl acrylate was used as the second monomer for the synthesis of the block copolymer (Supplementary Scheme 1). After chain extension, a clear shift to high-molecular-weight region ($M_n = 30,100$) and a narrow molecular weight distribution ($D = 1.15$) were observed in GPC traces (Fig. R4), suggesting the high chain-end fidelity of *Pn*BA synthesized via the aerobic mechano-RAFT polymerization.

End-group analysis was further conducted via MALDI-TOF mass spectroscopy. The strongest peak revealed a molecular weight of 8,054 ($DP = 61$, Fig. R5a and Supplementary Fig. 13) which matched well with the corresponding GPC trace ($M_n = 7,500$, Fig. R5b). Another distribution at 8,069 can be attributed to the removal of a methyl group by laser irradiation, and the main peak with intervals of 128.03 Da corresponded to the molar mass of the *n*BA unit (Fig. R5a).

To clarify high end group fidelity of the aerobic mechano-RAFT polymerization, we made two figures that you will find below.

Figure R16. Chain extension of the *Pn*BA-*b*-EA from *Pn*BA by this aerobic mechano-RAFT.

Figure R17. PnBA synthesized via the aerobic mechano-RAFT. a) The MALDI-TOF spectrum; b) the GPC trace.

We added the figure to Figure 2 and revised following sentences to the main text (Results)

Results (Figure 2):

Fig. 2 Aerobic mechano-RAFT driven by ball-milling. **a** Schematic polymerization of nBA. **b** Physical pictures and infrared thermal images of the ball milling jar before and after reaction. **c** Controls for aerobic mechano-RAFT reaction conditions: [nBA]: [BTPA]: [Et₃B-PyOMe] = 200:1:5, 100 μL (0.03 mL/g) DMF as LAG, 35 mL zirconium oxide milling jar with four 8 mm diameter zirconium oxide balls, Reaction condition: 30 Hz-2 h, and conversion was determined by ¹H NMR

spectroscopy. **d** Kinetic plot evolution of mechano-RAFT under various ball milling frequencies. **e** The evolution of molecular weight and dispersity for the polymer versus conversion. **f** GPC traces of aerobic mechano-RAFT with different DP_T . **g** GPC traces of PnBA and chain-extended polymer.

“Chain extension was further conducted to examine the chain-end fidelity of the synthesized polymer by this approach. nBA was polymerized under identical conditions to give first block PnBA ($M_n = 14,900$ and $\mathcal{D} = 1.15$). Ethyl acrylate was used as the second monomer for the synthesis of the block copolymer (Supplementary Scheme 1). After chain extension, a clear shift to high-molecular-weight region ($M_n = 30,100$) and a narrow molecular weight distribution ($\mathcal{D} = 1.15$) were observed in GPC traces (Fig. 2g), suggesting the high chain-end fidelity of PnBA synthesized via the aerobic mechano-RAFT polymerization.”

“Another feature of this aerobic mechano-RAFT is the high retention of chain end for the synthesized polymer. Matrix-assisted laser desorption/ionization-time of flight (MALDI-TOF) mass spectra of PnBA showed that the strongest peak revealed a molecular weight of 8,054 ($DP = 61$, Supplementary Fig. 13) which matched well with the corresponding GPC trace ($M_n = 7,500$, Supplementary Fig. 14). Another distribution at 8,069 can be attributed to the removal of a methyl group by laser irradiation³³, and the main peak with intervals of 128.03 corresponded to the molar mass of the nBA unit (Supplementary Fig. 15).”

Methods:

MALDI-TOF mass spectroscopy of PnBA. PnBA for MALDI-TOF mass spectroscopy were obtained from general aerobic mechano-RAFT procedure (30 Hz, 1h). MALDI-TOF mass spectrometer was from Bruker, Germany. The MALDI instrument was equipped with a 337 nm pulsed nitrogen laser (laser intensity of 50 Hz). The number of laser irradiations was 100 for all mass spectra (delay time of 190 ns), with a 20 kV acceleration voltage. MALDI experiment was carried out using 2,5-DHB as the matrix. The matrix solution was prepared by dissolving 40 mg of 2,5-DHB in 1 mL of THF.”

Comment 8. What is the kinetics of Et₃B release/free from the complex? How does this release kinetics relate to the kinetics of radical formation and polymerization?

Response: Thanks for your insightful comments. The kinetics of Et₃B release/free from the complex could hardly be determined throughout ball milling. However, the dosage of the mechano-labile initiators (Supplementary Table 1) was screened out to optimize reaction conditions. As shown in the Table S1, the polymerization rate significantly increased as the ratio of CTA: Et₃B-PyOM decreased from 1:1 to 1:5, and then slowed down as the ratio further decreased.

Table R2 Aerobic mechano-RAFT with different initiator dosages

Entry ^[a]	Monomer	[CTA]:[I]	Conversion ^[b]	$M_{n,th}$	$M_{n,GPC}$	$\mathcal{D}^{[c]}$
1	n BA	1:1	<11%	/	/	/
2	n BA	1:3	30%	7900	18900	1.08
3	n BA	1:5	69%	17900	22400	1.10
4	n BA	1:6	51%	13300	19400	1.09

Comment 9. The comparison of photo- and thermo-RAFT (Figure 4) without using the ball milling is also not convincing. The uniformity and luminescent properties may be a result of the well-known mixing effect of the ball milling. If this is the case, why not just use traditional RAFT polymerization with a good mixing approach such as ball milling, high-shear flow or ultrasonification? I recommend the authors perform control experiments of photo- and thermo-RAFT (Figure 4) using the ball milling.

Response: Thanks for the insightful comments. Ball milling is essential to achieve uniform and emissive polymer/perovskites hybrids. This aerobic mechanochemical approach is a solventless method to access polymers and polymer/inorganic hybrids, which complements the photochemical, electrochemical and thermochemical approaches. The photo- and thermo-RAFT using the ball milling will definitely complicate the equipment, and elevate the cost and energy consumption, at least for the synthesis of polymer/perovskites hybrids.

To better convey the advantage of this aerobic mechano-RDRP, we optimized the design for polymer/perovskites hybrids. NIPAM was selected as the functional monomer to coordinate and passivate perovskite nanocrystals (PNCs). As illustrated in **Fig.4a**, PNCs@P(*n*BA-*co*-NIPAM) were synthesized by one-pot ball milling (20 Hz with four 10 mm zirconia balls) without additional solvent and further purification. Well-defined polymers were achieved with high conversion (>80 %, Supplementary Fig.30) and narrow molecular weight distribution ($\mathcal{D} = 1.25$, Supplementary Fig.31). The signal intensity analysis of ¹³C NMR spectroscopy (DEPT135, CH₃/CH positive and CH₂ negative) shows that the copolymerization ratio of *n*BA and NIPAM segment is 9.7, which matched well with the feeding ratio (**Fig.4b** and Supplementary Fig.32). The photoluminescence (PL) spectrum of this hybrids was displayed in **Fig. 4c**, with a maximum emissive wavelength around 529 nm and a full width at half-maximum (FWHM) of ~25.2 nm. In addition, the XRD pattern of the hybrid matched the standard card of MAPbBr₃ nanocrystals (**Fig.4d**). “IFE” characters on the glass were made by the injection of PNCs @ P(*n*BA-*co*-NIPAM), affording strong green fluorescence emission under 365 nm ultraviolet irradiation (**Fig.4e**). These results demonstrated the exciting potential of this aerobic mechano-RAFT as an operationally simple, mild and solvent-free route to synthesize polymer/inorganic hybrids from commercial monomers and precursors.

Results (Figure 4):

Figure 4. Polymer/perovskites hybrids by aerobic mechano-RAFT.

-[R1] Imato, K. *et al.* Mechanophores with a Reversible Radical System and Freezing-Induced Mechanochemistry in Polymer Solutions and Gels. *Angewandte Chemie International Edition* **54**, 6168-6172 (2015).

-[R2] Hermansen, L., Hultman, E. & Saltin, B. Muscle Glycogen during Prolonged Severe Exercise. *Acta Physiologica Scandinavica* **71**, 129-139 (1967).

-[R3] VØLlestad, N. K. & Blom, P. C. S. Effect of varying exercise intensity on glycogen depletion in human muscle fibres. *Acta Physiologica Scandinavica* **125**, 395-405 (1985).

-[R4] Murray, B. & Rosenbloom, C. Fundamentals of glycogen metabolism for coaches and athletes. *Nutrition Reviews* **76**, 243-259 (2018).

* * * * *

Krzysztof Matyjaszewski
Department of Chemistry
Carnegie Mellon University
4400 Fifth Avenue, Pittsburgh, PA 15213, USA
Tel: + 1 412-268-3209
Email: matyjaszewski@cmu.edu

Xiangcheng Pan
State Key Laboratory of Molecular Engineering of Polymers
Department of Macromolecular Science
Fudan University, Shanghai 200438, China
Tel: + 86 21-31242898
Email: panxc@fudan.edu.cn

Zhenhua Wang
Institute of Flexible Electronics (IFE)
Northwestern Polytechnical University (NPU)
Xi'an 710072, China
Tel: + 86 29 8846 0624
Email: iamzhwang@nwpu.edu.cn

Reviewers' Comments:

Reviewer #1:

Remarks to the Author:

I am mostly satisfied with responses to my comments (original reviewer 1) but still have some questions remaining from my original comments and those from other reviewers. I am more in favor than the other two reviewers in terms of novelty/impact as previous ball-milling RDRP processes struggle with certain monomer classes that work in this case (Reviewer 2, Comment 4) and because a relatively novel non-mechanoredox (and air tolerant) initiation method is utilized.

Reviewer 1, Comment 1 + Reviewer 3: The authors suggest that mechanoredox polymerizations give rise to "the complicated purification procedure for polymer products". This is simply not the case and piezoelectric particles can easily be removed/reused. The authors propose a new initiation mechanisms in this work and have added a great deal of additional mechanistic studies. However selling their work on purification alone is unjustified in this case.

Reviewer 1, Comment 3: I appreciate the additional experiments and corresponding data added to the manuscript, but I am confused about where the control experiments went. My original comment referenced photo and thermal RDRP control experiments in the original manuscript. Those experiments seem to be gone now (?) and therefore my question wasn't really answered. I understand better how the mechano-RDRP process works, but how it compares to other technologies for making polymer/perovskite blends is still unclear to me.

Reviewer 1, Comment 6: This is a minor point, but I disagree that molar mass is dimensionless (regardless of literature cited). A calibration curve is generated with known polymer standards with a given Mn. Conventional GPC calculates Mn based on this calibration curve and therefore the corresponding units should also be g/mol or Da.

Reviewer #3:

Remarks to the Author:

I appreciate the author's efforts to address the reviewers' feedback. However, upon a thorough examination of both the response and the revised manuscript, I maintain my stand that this work is more suitable for publication in a specialized polymer journal, given its limited novelty and significance.

The authors commence the rebuttal with two extensive paragraphs explaining the significance and historical context of mechanochemistry and photocontrolled radical polymerization, which in my view is simply redundancy. Each study should be assessed independently, and many great works in one research area do not necessarily mean this manuscript has the same level of novelty and significance. I am fully aware of the contributions outlined in these two paragraphs, yet they are unrelated to this study. It is more important to discuss previous work similar to this study, as previously mentioned in my comments. However, it appears that the authors have overlooked prior contributions in this field and have not addressed my concerns regarding the similarities to these papers (Macromolecules 2007, 40, 10, 3554-3560 and Angew. Chem. 2018, 130, 9574-9577). For example, the argument that "mechano-RDRP based on the combinative attributes from molecular oxygen and mechanical forces has not yet been reported in the literature" lacks discussion in light of Fedorov and colleagues' work over a decade ago, wherein a similar alkylborane complex was utilized to initiate free radical polymerization, and the Pan group's study showing reaction between triethylborane and oxygen to facilitate similar RAFT polymerization. The contribution of this work to the field only lies in the release of Et3B by mechanical force rather than the mechano-RAFT polymerization.

In addition, I also find the majority of arguments presented in the rebuttal unconvincing. For instance, the authors comment that "We provide evidence of the ability of this organo-driven aerobic mechano-RDRP to polymerize a wide array of monomers including (meth)acrylates, styrenic monomers, and solid acrylamides at room temperature without additional solvents. This could hardly be achieved by the thermal or photo-initiation system." However, it is widely acknowledged that RAFT polymerization, including thermal and photocatalyzed systems, is compatible with a broad scope of monomers.

Therefore, this comment is not accurate and this work lacks novelty in expanding the monomer scope. Furthermore, open-to-air polymerization has been extensively documented in the literature (e.g., refer to this review *Chem. Soc. Rev.*, 2018,47, 4357-4387), yet the authors once again fail to acknowledge previous studies when making their claims. Solvent-less method, no need for purification, and metal-free polymerization should be also discussed in light of the current literature.

Regarding my previous detailed comments, it appears that the authors have chosen to overlook some of them. Specifically, I recommend the addition of an introduction to RAFT polymerization (comment 1). The claim regarding ATRP should either be demonstrated in this manuscript or removed altogether but the authors insist on making a claim based on future work (comment 2). Additionally, the comparison between aerobic mechano-RDRP in this work and skeletal muscle in aerobic exercise lacks scientific relevance and should be removed (comment 6).

Reviewer #4:

Remarks to the Author:

[Note from the Editor: Reviewer #4 was asked to look over the response given to reviewer #2 who was not able to look over the revision again.]

The authors have made significant improvements in revising their manuscript. In my opinion, this is excellent work that deserves publication. I have one minor suggestion regarding Figure 1. Although I understand the intention of providing a comparison with natural or biological processes, I believe that the comparison is somewhat overstretched. I suggest removing Figure 1 (top section) as this part is not needed.

Another minor comment concerns Figure 2f, where DP = 1000. The dispersity is a bit high for PBA. While the authors mentioned that it is lower than previous reports, those reports were for PMMA (see lines 110-114). This is not a fair comparison, as MMA typically yields higher dispersity than acrylates or acrylamides. I believe the slightly high dispersity in this case can be attributed to the presence of some oxygen, which may have terminated some polymer chains, or a relatively high concentration of radicals. Although this is a limitation, it does not diminish the overall impact of the paper.

Please check this sentence: "Matrix-assisted laser desorption/ionization-time of flight (MALDI-TOF) mass spectra of PnBA showed that the strongest peak revealed a molecular weight of 8,054 (DP = 61, Supplementary Fig. 13) which matched well with the corresponding GPC trace ($M_n = 7,500$, Supplementary Fig. 14)." I think the difference is significant and perhaps the authors should avoid use match well. Given that MALDI-TOF is not highly quantitative, the authors should consider using more precise language.

Additionally, as this study involves oxygen-tolerant polymerization, specifically RAFT, the authors should cite relevant references on this topic.

In conclusion, the authors have done an excellent job addressing the reviewers' comments carefully, and I believe this work will make a significant contribution to the field.

We warmly welcome the reviewers' comments and deeply appreciate the fact they have taken the time to examine the manuscript in great detail and provided many pertinent comments. The manuscript has been greatly improved following this round of review; we believe we have addressed the comments and concerns of the reviewers appropriately.

REVIEWER COMMENTS

Reviewer #1 (Remarks to the Author):

I am mostly satisfied with responses to my comments (original reviewer 1) but still have some questions remaining from my original comments and those from other reviewers. I am more in favor than the other two reviewers in terms of novelty/impact as previous ball-milling RDRP processes struggle with certain monomer classes that work in this case (Reviewer 2, Comment 4) and because a relatively novel non-mechanoredox (and air tolerant) initiation method is utilized.

We thank Reviewer #1 for reviewing our paper again and providing their positive feedbacks.

Comment 1: The authors suggest that mechanoredox polymerizations give rise to “the complicated purification procedure for polymer products”. This is simply not the case and piezoelectric particles can easily be removed/reused. The authors propose a new initiation mechanisms in this work and have added a great deal of additional mechanistic studies. However selling their work on purification alone is unjustified in this case.

Response: Thanks for the valuable suggestion. We have revised the descriptions to sell our work.

Introduction:

“Mechanically controlled radical polymerization relying on piezoelectricity, contact electrification or sonolysis require metal-based mechanotransducers or high-energy force to activate polymerization via mechano-electro-chemical transformation, constraining their applications particularly in biomedicine and electronics.”

Comment 2: I appreciate the additional experiments and corresponding data added to the manuscript, but I am confused about where the control experiments went. My original comment referenced photo and thermal RDRP control experiments in the original manuscript. Those experiments seem to be gone now (?) and therefore my question wasn't really answered. I understand better how the mechano-RDRP process works, but how it compares to other technologies for making polymer/perovskite blends is still unclear to me.

Response: Thank you for your insightful comments. In the first version, we tried to synthesize polymer/perovskites hybrids via mechano-, photo- and thermo- RDRP procedure without solvent for

comparison to present the advantage of mechanochemical approach. As expected, the polymer/perovskites hybrids obtained by mechano-RDRP revealed strong and uniform green fluorescence emission under 365 nm ultraviolet irradiation. Another two control experiments performed limited fluorescence emission under UV irradiation. In SEM images, the polymer/perovskites hybrids by mechano-RDRP procedure contained perovskites nanocrystals with diameter less than 50 nm, while the size of perovskite particles synthesized by the photo- or thermo-RDRP was in the range of several hundred nm to several micrometers. Due to concurrent quantum confinement and size dependent structural effects, smaller grains can spatially limit the exciton diffusion length or charge carriers and reduce the possibility of exciton dissociation into carriers, which caused substantial increases in steady-state photoluminescence intensity and efficiency of MAPbBr₃ particles (*Nature Nanotech* **9**, 687–692 (2014), *Science* **350**,1222-1225(2015)). These results presented that mechano-RDRP could be used to fabricate emissive polymer/perovskites hybrids due to the combination of mechanochemical effect for polymerization and shear effect for uniform perovskites nanocrystals.

Figure R1. The synthesis of polymer/perovskites hybrids was attempted via (a) mechano-, (b) photo- or (c) thermo- RDRP procedure

According to your insightful comment, in the second version, we optimized the design for the synthesis of polymer/perovskites hybrids by this approach without crosslinker so that well-defined copolymers could be fully characterized. As shown Supplementary Fig.31, the copolymer of *n*BA and NIPAM could be accessed with $M_n = 20,900$ Da and $M_w / M_n = 1.26$ via this mechanochemical approach. NIPAM was used as the functional monomer to passivate perovskite nanocrystals and strengthen the intermolecular interactions for the generation of perovskite nanocrystals.

Supplementary Fig. 31 GPC trace of the copolymerization of *n*BA and NIPAM based on aerobic mechano-RAFT

Comment 3: This is a minor point, but I disagree that molar mass is dimensionless (regardless of literature cited). A calibration curve is generated with known polymer standards with a given M_n . Conventional GPC calculates M_n based on this calibration curve and therefore the corresponding units should also be g/mol or Da.

Response: Thank you for the valuable suggestions. We have added the corresponding units in the text and all the figures.

Figure 2. Aerobic mechano-RAFT driven by ball-milling.

Results (Figure 2):

“Another feature of this aerobic mechano-RAFT is the high retention of chain end for the synthesized polymer. Matrix-assisted laser desorption/ionization-time of flight (MALDI-TOF) mass spectra of PnBA showed that the strongest peak revealed a molecular weight of 8,054 Da ($DP = 61$, Supplementary Fig. 13) which was closed with the corresponding GPC trace ($M_n = 7,500$ Da, Supplementary Fig. 14). Another distribution at 8,069 Da can be attributed to the removal of a methyl group by laser irradiation³³, and the main peak with intervals of 128.03 Da corresponded to the molar mass of the nBA unit (Supplementary Fig. 15). Chain extension was further conducted to examine the chain-end fidelity of the synthesized polymer by this approach. nBA was polymerized under identical conditions to give first block PnBA ($M_n = 14,900$ Da and $\bar{D} = 1.15$). Ethyl acrylate was used as the second monomer for the synthesis of the block copolymer (Supplementary Scheme 1). After chain extension, a clear shift to high-molecular-weight region ($M_n = 30,100$ Da) and a narrow molecular weight distribution ($\bar{D} = 1.15$) were observed in GPC traces (Fig. 2g), suggesting the high chain-end fidelity of PnBA synthesized via the aerobic mechano-RAFT polymerization.”

Table 1 Monomer scope for aerobic mechano-RAFT

Entry	Monomer	DP_T	Conv. (%) ^a	$M_{n,theo}$ (Da) ^b	$M_{n,GPC}$ (Da) ^c	\mathcal{D} ^c	I^* (%) ^d
1	n BA	200	81	21000	20800	1.11	>99
2	EA	200	75	15200	13400	1.14	>99
3	t BA	200	65	16900	12600	1.16	>99
4	MA	200	65	11400	15100	1.09	76
5	MMA	200	32	6600	8900	1.50	74
6	St	200	32	7100	6100	1.22	>99
7	VN	100	30	4800	4400	1.21	>99
8	VC	100	25	5050	5400	1.45	>94
9	NIPAM	100	>95	11600	25800	1.36	45
10	NPA	100	>95	15000	18900	1.19	79

^a Conversion was determined by ¹H NMR spectroscopy. ^b $M_{n,theo} = M_{end\ group} + [M]_0/[CTA]_0 \times conversion$
 $\times M_{monomer}$. ^c M_n and M_w/M_n were determined by GPC. ^d Initiator efficiency (I^*) = $M_{n,theo}/M_{n,exp} \times 100$.

Results (Table 1):

“Well-defined polymers were achieved for all the acrylates, but the polymerization of MMA revealed a wider molecular weight distribution ($\mathcal{D} = 1.50$) due to the slow initiating rate of BTPA for methacrylates (Supplementary Fig.17-19). Then, 4-(benzenecarbonothioylsulfanyl)-4-cyanopentanoic acid (BTCPA) was selected as the CTA for the polymerization of MMA with 10 mm balls and an extended reaction time (5 h), afford a higher conversion (~52%, Supplementary Fig.20) and well-

defined PMMA ($M_n = 8,300$ Da, $\bar{D} = 1.19$, Supplementary Fig.21).”

Supplementary Information:

Supplementary Fig. 5 GPC traces for the polymerization of nBA under 30 Hz ball milling

Supplementary Fig. 7 GPC traces for the polymerization of nBA under 10 Hz ball milling

Supplementary Fig. 9 GPC traces for the polymerization of nBA under 20 Hz ball milling

Supplementary Fig. 14 GPC trace of PnBA

Supplementary Fig. 21 GPC traces for the polymerization of MMA via Process 2

Supplementary Fig. 29 GPC traces (in DMF solution) of NIPAM with different DP_T via Process 3

Supplementary Fig. 31 GPC trace of the copolymerization of *n*BA and NIPAM based on aerobic mechano-RAFT

Supplementary Table 1 Aerobic mechano-RAFT with different initiator dosages

Entry ^[a]	Monomer	[CTA]:[I]	Conversion ^[b]	$M_{n,th}$ (Da)	$M_{n,GPC}$ (Da)	D ^[c]
1	n BA	1:1	<11%	/	/	/
2	n BA	1:3	30%	7900	18900	1.08
3	n BA	1:5	69%	17900	22400	1.10
4	n BA	1:6	51%	13300	19400	1.09

^a Reaction conditions: [M]:[BTPA]:[Et₃B-PyOMe]=200:1:X, no LAG, ball milling (35 mL zirconium oxide jar, 8 mm zirconium oxide grinding ball, 30 Hz), Reaction time-2 h. ^b Conversion was determined by ¹H NMR spectroscopy. ^c M_n and D were determined by GPC.

Supplementary Table 2 Aerobic mechano-RAFT with different LAG

Entry ^[a]	Monomer	LAG	Conversion ^[b]	$M_{n,th}$ (Da)	$M_{n,GPC}$ (Da)	D ^[c]
1	n BA	NO	69%	17900	22400	1.10
2	n BA	DMF	81%	21000	20800	1.11
3	n BA	DMSO	76%	19700	4760	3.08
4	n BA	Acetone	77%	20000	21500	1.12
5	n BA	Dioxane	81%	21000	23000	1.12

6 *n*BA MeCN 76% 19700 21800 1.09

^a Reaction conditions: [M]:[BTPA]:[Et₃B-PyOMe]=200:1:5, 100 μL (0.03 mL/g) as LAG, ball milling (35 mL zirconium oxide jar, 8 mm zirconium oxide grinding ball, 30 Hz), Reaction time-2 h. ^b Conversion was determined by ¹H NMR spectroscopy. ^c *M_n* and *D* were determined by GPC.

Supplementary Table 3 Aerobic mechano-RAFT with different CTAs

Entry ^[a]	Monomer	CTA	Conversion ^[b]	M_n ,th (Da)	M_n ,GPC (Da)	D ^[c]
1	n BA	BTPA	81%	21000	20800	1.11
2	n BA	ATTC	89%	23000	18100	1.16
3	n BA	BTCPA	13%	3600	1900	2.27
4	n BA	BDPA	92%	23800	89000	1.79

^a Reaction conditions: [M]:[CTA]:[Et₃B-PyOMe]=200:1:5, 100 μL (0.03 mL/g) DMF as LAG, ball milling (35 mL zirconium oxide jar, 8 mm zirconium oxide grinding ball, 30 Hz), Reaction time-2 h. ^b Conversion was determined by ¹H NMR spectroscopy. ^c *M_n* and *D* were determined by GPC.

Supplementary Table 4 Aerobic mechano-RAFT with different milling balls

Entry ^[a]	Monomer	Milling ball	Conversion ^[b]	M_n ,th (Da)	M_n ,GPC (Da)	D ^[c]
1	n BA	4*8mm	48%	12500	16500	1.10
2	n BA	6*8mm	53%	13800	20100	1.11
3	n BA	4*10mm	52%	13600	21000	1.09

^a Reaction conditions: [M]:[BTPA]:[Et₃B-PyOMe]=200:1:5, 100 μL (0.03 mL/g) DMF as LAG, ball milling (35 mL zirconium oxide jar, zirconium oxide grinding ball, 30 Hz), Reaction time-2 h. ^b Conversion was determined by ¹H NMR spectroscopy. ^c *M_n* and *D* were determined by GPC.

Supplementary Table 5 Aerobic mechano-RAFT with different ambient temperature

Entry ^[a]	Monomer	Temperature/°C	Conversion ^[b]	M_n ,th (Da)	M_n ,GPC (Da)	D ^[c]
1	n BA	20	65%	16900	16900	1.09
2	n BA	25	65%	16900	20800	1.11
3	n BA	30	89%	23000	29900	1.09

^a Reaction conditions: [M]:[BTPA]:[Et₃B-PyOMe]=200:1:5, 100 μL (0.03 mL/g) DMF as LAG, ball milling (35 mL zirconium oxide jar, zirconium oxide grinding ball, 30 Hz), Reaction time-2 h. ^b Conversion was determined by ¹H NMR spectroscopy. ^c *M_n* and *D* were determined by GPC.

Supplementary Table 6 Aerobic mechano-RAFT with different air volume

Entry ^[a]	Monomer	Jar	Conversion ^[b]	M_{n,th} (Da)	M_{n,GPC} (Da)	D ^[c]
1	n BA	35 ml	81%	21000	20800	1.11
2	n BA	25 ml	81%	21000	20500	1.11

^a Reaction conditions: [M]:[BTPA]:[Et₃B-PyOMe]=200:1:5, 100 μL (0.03 mL/g) as LAG, ball milling (X mL zirconium oxide jar, 8 mm zirconium oxide grinding ball, 30 Hz), Reaction time-2 h. ^b Conversion was determined by ¹H NMR spectroscopy. ^c *M_n* and *D* were determined by GPC.

Reviewer #3 (Remarks to the Author):

I appreciate the author's efforts to address the reviewers' feedback. However, upon a thorough examination of both the response and the revised manuscript, I maintain my stand that this work is more suitable for publication in a specialized polymer journal, given its limited novelty and significance. The authors commence the rebuttal with two extensive paragraphs explaining the significance and historical context of mechanochemistry and photocontrolled radical polymerization, which in my view is simply redundancy. Each study should be assessed independently, and many great works in one research area do not necessarily mean this manuscript has the same level of novelty and significance. I am fully aware of the contributions outlined in these two paragraphs, yet they are unrelated to this study. It is more important to discuss previous work similar to this study, as previously mentioned in my comments.

Response: Thank you for highlighting that we did not properly present the novelty of our current work in the previous versions of the manuscript. We are confident we addressed your doubts appropriately and improved the manuscript based on your comments.

Comment 1: However, it appears that the authors have overlooked prior contributions in this field and have not addressed my concerns regarding the similarities to these papers (*Macromolecules* 2007, 40, 10, 3554-3560 and *Angew. Chem.* 2018, 130, 9574-9577). For example, the argument that "mechano-RDRP based on the combinative attributes from molecular oxygen and mechanical forces has not yet been reported in the literature" lacks discussion in light of Fedorov and colleagues' work over a decade ago, wherein a similar alkylborane complex was utilized to initiate free radical polymerization, and the Pan group's study showing reaction between triethylborane and oxygen to facilitate similar RAFT polymerization. The contribution of this work to the field only lies in the release of Et₃B by mechanical force rather than the mechano-RAFT polymerization. In addition, I also find the majority of arguments presented in the rebuttal unconvincing. For instance, the authors comment that "We provide evidence of the ability of this organo-driven aerobic mechano-RDRP to polymerize a wide array of monomers including (meth)acrylates, styrenic monomers, and solid acrylamides at room temperature without additional solvents. This could hardly be achieved by the thermal or photo-initiation system." However, it is widely acknowledged that RAFT polymerization, including thermal and photocatalyzed systems, is compatible with a broad scope of monomers. Therefore, this comment is not accurate and this work lacks novelty in expanding the monomer scope.

Response: Thank you for your insightful comments and recognizing our previous work. Our current study lies in the background that mechanically controlled polymerization or polymer crosslinking are of particular interest for designing smart materials that strengthen under stresses commonly encountered in engineering material (*Science* 363,504-508(2019), *Nat. Mater.* 20, 869-874 (2021)). Previous papers by Fedorov and colleagues proposed elegant approaches to solving the problem of oxygen inhibition for radical polymerization using alkylborane complex (*Macromolecules* 40, 10, 3554-3560(2007), *Macromolecules* 39, 17, 5669-5674(2006)). However, there was no mechanical regulation involved in these papers. In term of our work (*Angew. Chem.*130, 9574-9577(2018)), no mechanical regulation was reported either.

In addition, mechanically controlled polymerization relying on piezoelectricity (*Nat. Chem.* **9**, 135-139 (2017)), contact electrification (*Angew. Chem. Int. Ed.* **62**, e202215733 (2023)) or sonolysis (*Angew. Chem. Int. Ed.* **56**, 12302 (2017)) require metal-based mechanotransducers or high-energy force to activate polymerization via mechano-electro-chemical transformation, constraining their applications particularly in biomedicine and electronics. In current study, we proposed a mechanistically distinct approach taking advantage of organic mechano-labile complexes that could be activated by ball milling to release reactive species required for the oxidative process to produce initiating radicals for polymerization of vinyl monomers to generate well-defined polymers. Compared to the previous mechanochemical systems, this approach features the following merits: 1) in-air polymerization without degassing; 2) low-energy mechanical activation for minimized side reactions to access high MW polymer; 3) minimized metal contamination and the avoidance of potentially harmful organic solvents; 4) expanded monomer scope including solid monomers as well as the bulk synthesis of polymer/perovskite hybrids without solvents at room temperature. This approach also complements the thermo- or photo-controlled systems which are challenging to access solid polymerization or polymer hybrids in a controlled manner without solvents.

To take into account your remark and convey better the advancement, we presented a figure as below to show the mechanochemical polymerization of solid monomers without solvent in a controlled manner.

Monomer	a		b	
				
Time, Conv.	5h, 95%	5h, 95%	5h, 30%	5h, 25%
$M_{n,GPC}(Da)$	25 800	18 900	4 400	5 400
\bar{D}	1.36	1.19	1.21	1.45

^a Reaction condition: in 25 mL zirconium oxide jar equipped with four 10 mm zirconium oxide balls, without solvent and no liquid assisted grinding, ball-milled at 20 Hz for 5 h. ^b Reaction condition: in 25 mL stainless steel jar equipped with four 10 mm stainless steel balls, without solvent and no liquid assisted grinding, ball-milled at 30 Hz for 5 h.

Figure R2. The solid monomer scope for the aerobic mechanochemical RAFT

Figure R3. GPC traces of acrylamide monomers

Figure R4. GPC traces of poly(vinyl naphthalene) (PVN) and poly(vinyl carbazole) (PVC)

We also added the citations as follows:

- [53] Fedorov, A. V., Ermoshkin, A. A., Mejiritski, A. & Neckers, D. C. New Method To Reduce Oxygen Surface Inhibition by Photorelease of Boranes from Borane/Amine Complexes. *Macromolecules* **40**, 3554-3560 (2007).
- [54] Wilson, O. R. & Magenau, A. J. D. Oxygen Tolerant and Room Temperature RAFT through Alkylborane Initiation. *ACS Macro Letters* **7**, 370-375 (2018).

Results (Figure 1):

“Inspired by this, we designed a triethylborane/pyridine (Et₃B-Py) complex as the latent initiator consisting of triethylborane and an electron-donating pyridine for the mechanochemical conversion of oxygen into activators to enable RDRP. This design arose from previous findings that triethylborane

could react with molecular oxygen in air to form initiating radicals for polymerization^{39,50,51}.”

Comment 2: Furthermore, open-to-air polymerization has been extensively documented in the literature (e.g., refer to this review Chem. Soc. Rev., 2018,47, 4357-4387), yet the authors once again fail to acknowledge previous studies when making their claims. Solvent-less method, no need for purification, and metal-free polymerization should be also discussed in light of the current literature.

Response: Thank you for the suggestion. You also recognized the enormous effort to access open-to-air polymerization in the past years. However, another fact is that the reported strategy for the removal of dissolved oxygen could hardly be compatible with mechanochemical radical polymerization. There was no paper published on this topic. This is why we conducted this work.

Introduction:

“Recently, enormous effort has been devoted to the removal of dissolved oxygen prior to polymerization, including approaches employing enzymes^{8,9,39}, microbial metabolisms^{7,10,11}, reducing agents⁴⁰⁻⁴⁴ or photocatalysts⁴⁵⁻⁴⁷. The broad success of oxygen-tolerant RDRP hinges on the susceptibility of the chemical conversion of molecular oxygen with high levels of efficiency and selectivity⁴⁸.”

Results (Figure 1):

“Inspired by this, we designed a triethylborane/pyridine (Et₃B-Py) complex as the latent initiator consisting of triethylborane and an electron-donating pyridine for the mechanochemical conversion of oxygen into activators to enable RDRP. This design arose from previous findings that triethylborane could react with molecular oxygen in air to form initiating radicals for polymerization^{39,50,51}.”

*“Furthermore, we tried to synthesize a polymer with $DP_T = 1,000$ and obtained a PnBA with an actual $DP = 950$ and $\bar{D} = 1.29$ through the optimized method (Supplementary Fig. 12, details in **Methods** section). We believe the slightly high dispersity in this case can be attributed to the presence of some oxygen, which may have terminated some polymer chains, or a relatively high concentration of radicals⁴⁸.”*

- [39] Chapman, R., Gormley, A. J., Herpoldt, K.-L. & Stevens, M. M. Highly Controlled Open Vessel RAFT Polymerizations by Enzyme Degassing. *Macromolecules* **47**, 8541-8547 (2014).
- [44] Vandenberg, J., Schweitzer-Chaput, B., Klusmann, M. & Junkers, T. Acid-Induced Room Temperature RAFT Polymerization: Synthesis and Mechanistic Insights. *Macromolecules* **49**, 4124-4135 (2016).
- [48] Yeow, J., Chapman, R., Gormley, A. J. & Boyer, C. Up in the air: oxygen tolerance in controlled/living radical polymerisation. *Chemical Society Reviews* **47**, 4357-4387 (2018).
- [53] Fedorov, A. V., Ermoshkin, A. A., Mejiritski, A. & Neckers, D. C. New Method To Reduce Oxygen Surface Inhibition by Photorelease of Boranes from Borane/Amine Complexes. *Macromolecules* **40**, 3554-3560 (2007).

[54] Wilson, O. R. & Magenau, A. J. D. Oxygen Tolerant and Room Temperature RAFT through Alkylborane Initiation. *ACS Macro Letters* **7**, 370-375 (2018).

Comment 4: Regarding my previous detailed comments, it appears that the authors have chosen to overlook some of them. Specifically, I recommend the addition of an introduction to RAFT polymerization (comment 1). The claim regarding ATRP should either be demonstrated in this manuscript or removed altogether but the authors insist on making a claim based on future work (comment 2).

Response: Thank you for the suggestion. We have mentioned in the manuscript that reversible addition–fragmentation chain transfer polymerization as the model method to verify the feasibility of the aerobic mechanochemical concept. First, we didn't include the part of ATRP in current study because the content will be over saturated. Second, we are going to conduct the aerobic mechano-ATRP to fabricate functional hybrids using ultrasound. Based on your request, we presented the example of ATRP of MA as below. Although the dispersity is relatively high in the ultrasonic system, we believe we can optimize this in the future as excellent control over polymerization was achieved in the optimized thermal system (*unpublished results*).

Table R1 Aerobic mechano-ATRP under ultrasound

Entry ^[a]	Monomer	Conversion ^[b]	$M_{n,th}$ (Da)	$M_{n,GPC}$ (Da)	D ^[c]
1	MA	68%	5700	9300	1.71

^a [M]:[EBiB]:[CuBr₂]:[TPMA]:[Et₃B-PyOMe]=100:1:0.03:0.12:3, Ultrasonic (V[M]:V[DMF]=1:1, 45 kHz, at 25 °C), Reaction time-4

h. ^b Conversion was determined by ¹H NMR spectroscopy. ^c M_n and D were determined by GPC.

Figure R5. GPC trace of PMA

Figure R6. ^1H NMR spectra of MA monomers after ATRP under ultrasound

ATRP of EGMEA in the absence of external deoxygenation

Oxygen-tolerant ATRP of EGMEA targeting a range of DP_n

[EGMEA]/[EBiB]/[TEB-DMAP]/[Cu]/[Me ₆ TREN]	DP	Time (h)	Conv. (%)	$M_{n,th}$	$M_{n,GPC}$	D
50/1/5/0.005/0.02	50	1.5	94	6300	7300	1.24
100/1/5/0.01/0.04	100	1.5	91	12000	14100	1.14
200/1/5/0.02/0.08	200	1.5	93	24400	26500	1.15
400/1/5/0.04/0.16	400	2	91	47500	49400	1.19
600/1/5/0.06/0.24	600	2	91	71200	73700	1.19
800/1/5/0.08/0.32	800	2	89	92800	86900	1.18
1000/1/5/0.1/0.4	1000	2	87	113300	108600	1.18

Figure R7. Oxygen-tolerant ATRP of EGMEA targeting a range of DP_n

Comment 5. Additionally, the comparison between aerobic mechano-RDRP in this work and skeletal muscle in aerobic exercise lacks scientific relevance and should be removed (comment 6).

Response: Thank you for your insightful comments. We have modified figure 1, and you will find below, and in the revised manuscript.

Aerobic Mechanochemical RDRP based on organic mechano-labile complexes

Figure 1 Illustrative design of aerobic mechano-RDRP.

Reviewer #4 (Remarks to the Author):

[Note from the Editor: Reviewer #4 was asked to look over the response given to reviewer #2 who was not able to look over the revision again.]

The authors have made significant improvements in revising their manuscript. In my opinion, this is excellent work that deserves publication.

We thank Reviewer #4 for reviewing our paper and providing their positive feedbacks.

Comment 1: I have one minor suggestion regarding Figure 1. Although I understand the intention of providing a comparison with natural or biological processes, I believe that the comparison is somewhat overstretched. I suggest removing Figure 1 (top section) as this part is not needed.

Response: Thank you for your insightful comments. We have modified figure 1, and you will find below, and in the revised manuscript.

Aerobic Mechanochemical RDRP based on organic mechano-labile complexes

Figure 1 Illustrative design of aerobic mechano-RDRP.

Comment 2: Another minor comment concerns Figure 2f, where $DP = 1000$. The dispersity is a bit high for PBA. While the authors mentioned that it is lower than previous reports, those reports were for PMMA (see lines 110-114). This is not a fair comparison, as MMA typically yields higher dispersity than acrylates or acrylamides. I believe the slightly high dispersity in this case can be attributed to the presence of some oxygen, which may have terminated some polymer chains, or a relatively high concentration of radicals. Although this is a limitation, it does not diminish the overall impact of the paper.

Response: Thank you for your insightful comments and for highlighting that we did not properly cite reference. We modified this sentence, and you will find below and in the revised manuscript.

Results (Figure 2):

*“Furthermore, we tried to synthesize a polymer with $DP_T = 1,000$ and obtained a PnBA with an actual $DP = 950$ and $\bar{D} = 1.29$ through the optimized method (Supplementary Fig. 12, details in **Methods** section). The slightly high dispersity in this case is potentially attributed to the presence of some oxygen, which may terminated some polymer chains, or a relatively high concentration of radicals⁴⁸.”*

Comment 3: Please check this sentence: "Matrix-assisted laser desorption/ionization-time of flight (MALDI-TOF) mass spectra of PnBA showed that the strongest peak revealed a molecular weight of 8,054 ($DP = 61$, Supplementary Fig. 13) which matched well with the corresponding GPC trace ($M_n = 7,500$, Supplementary Fig. 14)." I think the difference is significant and perhaps the authors should avoid use match well. Given that MALDI-TOF is not highly quantitative, the authors should consider using more precise language.

Response: Thank you for your valuable comments. We have modified this sentence, and you will find below and in the revised manuscript.

Results (Figure 2):

“Another feature of this aerobic mechano-RAFT is the high retention of chain end for the synthesized polymer. Matrix-assisted laser desorption/ionization-time of flight (MALDI-TOF) mass spectra of PnBA showed that the strongest peak revealed a molecular weight of 8,054 Da ($DP = 61$, Supplementary Fig. 13) which was closed to the corresponding GPC trace ($M_n = 7,500$ Da, Supplementary Fig. 14).”

Comment 4: Additionally, as this study involves oxygen-tolerant polymerization, specifically RAFT, the authors should cite relevant references on this topic.

Response: Thank you for the suggestion. We cite references about oxygen tolerate RAFT and you will find below and in the revised manuscript.

Introduction:

“Recently, enormous effort has been devoted to the removal of dissolved oxygen prior to polymerization, including approaches employing enzymes^{8,9,39}, microbial metabolisms^{7,10,11}, reducing agents⁴⁰⁻⁴⁴ or photocatalysts⁴⁵⁻⁴⁷. The broad success of oxygen-tolerant RDRP hinges on the susceptibility of the chemical conversion of molecular oxygen with high levels of efficiency and selectivity⁴⁸.”

Results (Figure 1):

“Inspired by this, we designed a triethylborane/pyridine (Et_3B -Py) complex as the latent initiator consisting of triethylborane and an electron-donating pyridine for the mechanochemical conversion of oxygen into activators to enable RDRP. This design arose from previous findings that triethylborane could react with molecular oxygen in air to form initiating radicals for polymerization^{39,50,51}.”

“Furthermore, we tried to synthesize a polymer with $DP_T = 1,000$ and obtained a PnBA with an actual $DP = 950$ and $\mathcal{D} = 1.29$ through the optimized method (Supplementary Fig. 12, details in **Methods** section). We believe the slightly high dispersity in this case can be attributed to the presence of some oxygen, which may have terminated some polymer chains, or a relatively high concentration of radicals⁴⁸.”

- [39] Chapman, R., Gormley, A. J., Herpoldt, K.-L. & Stevens, M. M. Highly Controlled Open Vessel RAFT Polymerizations by Enzyme Degassing. *Macromolecules* **47**, 8541-8547 (2014).
- [44] Vandenberg, J., Schweitzer-Chaput, B., Klussmann, M. & Junkers, T. Acid-Induced Room Temperature RAFT Polymerization: Synthesis and Mechanistic Insights. *Macromolecules* **49**, 4124-4135 (2016).
- [48] Yeow, J., Chapman, R., Gormley, A. J. & Boyer, C. Up in the air: oxygen tolerance in controlled/living radical polymerisation. *Chemical Society Reviews* **47**, 4357-4387 (2018).
- [53] Fedorov, A. V., Ermoshkin, A. A., Mejiritski, A. & Neckers, D. C. New Method To Reduce Oxygen Surface Inhibition by Photorelease of Boranes from Borane/Amine Complexes. *Macromolecules* **40**, 3554-3560 (2007).
- [54] Wilson, O. R. & Magenau, A. J. D. Oxygen Tolerant and Room Temperature RAFT through Alkylborane Initiation. *ACS Macro Letters* **7**, 370-375 (2018).

* * * * *

Reviewers' Comments:

Reviewer #1:

Remarks to the Author:

I am satisfied with the revised manuscript and recommend publication as is.